# A resource-rational theory of set size effects in human visual working memory

**Ronald van den Berg[1]\*, Wei Ji Ma[2]\***

[1]Department of Psychology, University of Uppsala, Uppsala, Sweden; [2]Center for Neural Science and Department of Psychology, New York University, New York, United States

**Abstract** Encoding precision in visual working memory decreases with the number of encoded items. Here, we propose a normative theory for such set size effects: the brain minimizes a weighted sum of an error-based behavioral cost and a neural encoding cost. We construct a model from this theory and find that it predicts set size effects. Notably, these effects are mediated by probing probability, which aligns with previous empirical findings. The model accounts well for effects of both set size and probing probability on encoding precision in nine delayed-estimation experiments. Moreover, we find support for the prediction that the total amount of invested resource can vary non-monotonically with set size. Finally, we show that it is sometimes optimal to encode only a subset or even none of the relevant items in a task. Our findings raise the possibility that cognitive "limitations" arise from rational cost minimization rather than from constraints.
DOI: https://doi.org/10.7554/eLife.34963.001

## Introduction

A well-established property of visual working memory (VWM) is that the precision with which items are encoded decreases with the number of encoded items (*Ma et al., 2014*; *Luck and Vogel, 2013*). A common way to explain this set size effect has been to assume that there is a fixed amount of resource available for encoding: the more items, the less resource per item and, therefore, the lower the precision per item. Different forms have been proposed for this encoding resource, such as samples (*Palmer, 1994*; *Sewell et al., 2014*), Fisher information (*van den Berg et al., 2012*; *Keshvari et al., 2013*), and neural firing rate (*Bays, 2014*). Models with a fixed amount of resource generally predict that the encoding precision per item (defined as inverse variance of the encoding error) is inversely proportional to set size. This prediction is often inconsistent with empirical data, which is the reason that more recent studies instead use a power law to describe set size effects (*Bays et al., 2009*; *Bays and Husain, 2008*; *van den Berg et al., 2012*; *van den Berg et al., 2014*; *Devkar et al., 2015*; *Elmore et al., 2011*; *Mazyar et al., 2012*; *Wilken and Ma, 2004*; *Donkin et al., 2016*; *Keshvari et al., 2013*). In these power-law models, the total amount of resource across all items is no longer fixed, but instead decreases or increases monotonically with set size. These models tend to provide excellent fits to experimental data, but they have been criticized for lacking a principled motivation (*Oberauer et al., 2016*; *Oberauer and Lin, 2017*): they accurately describe *how* memory precision depends on set size, but not *why* these effects are best described by a power law – or why they exist at all. In the present study, we seek a normative answer to these fundamental questions.

While previous studies have used normative theories to account for certain aspects of VWM, none of them has accounted for set size effects in a principled way. Examples include our own previous work on change detection (*Keshvari et al., 2012*; *Keshvari et al., 2013*), change localization (*van den Berg et al., 2012*), and visual search (*Mazyar et al., 2012*). In those studies, we modelled the decision stage using optimal-observer theory, but assumed an ad hoc power law to model the

**\*For correspondence:**
ronald.vandenberg@psyk.uu.se
(RB);
weijima@nyu.edu (WJM)

**Competing interests:** The authors declare that no competing interests exist.

**eLife digest** You can read this sentence from beginning to end without losing track of its meaning thanks to your working memory. This system temporarily stores information relevant to whatever task you are currently performing. However, the more items you try to hold in working memory at once, the poorer the quality of each of the resulting memories.

It has long been argued that this phenomenon – known as the set size effect – occurs because the brain devotes a fixed amount of neural resources to working memory. But this theory struggles to account for certain experimental results. It also fails to explain why the brain would not simply recruit more resources whenever it has more items to remember. After all, your heart does something similar by beating faster whenever you increase your physical activity.

Van den Berg and Ma break with the idea that working memory resources are fixed. They propose that resource allocation is flexible and driven by two conflicting goals: maximize memory performance, but use as few neural resources as necessary. Indeed, a computer simulation that follows this strategy mimics the set size effects seen in healthy volunteers. In the model, the items most relevant for a task are stored more accurately than less important ones, a phenomenon also observed in participants. Lastly, the simulation predicts that the total amount of resources devoted to working memory will vary with the number of items to be remembered. This too is consistent with the results of previous experiments.

Working memory thus appears to be more flexible than previously thought. The amount of resources that the brain allocates to working memory is not fixed but could be the result of balancing resource cost against cognitive performance. If this is confirmed, it may be possible to improve working memory by offering rewards, or by increasing the perceived importance of a task.
DOI: https://doi.org/10.7554/eLife.34963.002

relation between encoding precision and set size. Another example is the work by Sims and colleagues, who developed a normative framework in which working memory is conceptualized as an optimally performing information channel (*Sims, 2016*; *Sims et al., 2012*). Their information-theoretic framework offers parsimonious explanations for the relation between stimulus variability and encoding precision (*Sims et al., 2012*) and the non-Gaussian shape of encoding noise (*Sims, 2015*). However, it does not offer a normative explanation of set size effects. In their early work (*Sims et al., 2012*), they accounted for these effects by assuming that total information capacity is fixed, which is similar to other fixed-resource models and predicts an inverse proportionality between encoding precision and set size. In their later work (*Orhan et al., 2014*; *Sims, 2016*), they add to this the assumption that there is an inefficiency in distributing capacity across items and fit capacity as a free parameter at each set size. Neither of these assumptions has a normative motivation. Finally, Nassar and colleagues have proposed a normative model in which a strategic trade-off is made between the number of encoded items and their precision: when two items are very similar, they are encoded as a single item, such that there is more resource available per encoded item (*Nassar et al., 2018*). They showed that this kind of "chunking" is rational from an information-theoretical perspective, because it minimizes the observer's expected estimation error. However, just as in much of the work discussed above, this theory assumes a fixed resource budget for item encoding, which is not necessarily optimal when resource usage is costly.

The approach that we take here aligns with the recent proposal that cognitive systems are "resource-rational," that is, trade off the cost of using resources against expected task performance (*Griffiths et al., 2015*). The starting point of our theory is the principle that neural coding is costly (*Attwell and Laughlin, 2001*; *Lennie, 2003*; *Sterling and Laughlin, 2015*), which may have pressured the brain to trade off the behavioral benefits of high precision against the cost of the resource invested in stimulus encoding (*Pestilli and Carrasco, 2005*; *Lennie, 2003*; *Ma and Huang, 2009*; *Christie and Schrater, 2015*). We hypothesize that set size effects – and limitations in VWM in general – may be the result of making this trade-off near-optimally. We next formalize this hypothesis in a general model that can be applied to a broad range of tasks, analyze the theoretical predictions of this model, and fit it to data from nine previous delayed-estimation experiments.

## Theory

### General theoretical framework: trade-off between behavioral and neural cost

We define a vector $\mathbf{Q}=\{Q_1,\ldots,Q_N\}$ that specifies the amount of resource with which each of $N$ task-relevant items is encoded. We postulate that $\mathbf{Q}$ affects two types of cost: an expected behavioral cost $\bar{C}_{\text{neural}}(\mathbf{Q})$ induced by task errors and an expected neural cost $\bar{C}_{\text{neural}}(\mathbf{Q})$ induced by spending neural resources on encoding. The *expected total cost* is a weighted combination,

$$\bar{C}_{\text{total}}(\mathbf{Q};\lambda) = \bar{C}_{\text{behavioral}}(\mathbf{Q}) + \lambda \bar{C}_{\text{neural}}(\mathbf{Q}), \tag{1}$$

where the weight $\lambda \geq 0$ represents the importance of the neural cost relative to the behavioral cost. Generally, increasing the amount of resource spent on encoding will reduce the expected behavioral cost, but simultaneously increase the expected neural cost.

The key novelty of our theory is that instead of assuming that there is a fixed resource budget for stimulus encoding (a hard constraint), we postulate that the brain – possibly on a trial-by-trial basis – chooses its resource vector $\mathbf{Q}$ in a manner that minimizes the expected total cost. We denote the vector that yields this minimum by $\mathbf{Q}_{\text{optimal}}$:

$$\mathbf{Q}_{\text{optimal}} = \underset{\mathbf{Q}}{\text{argmin}}\, \bar{C}_{\text{total}}(\mathbf{Q};\lambda). \tag{2}$$

Under this policy, the total amount of invested resource – the sum of the elements of $\mathbf{Q}_{\text{optimal}}$ – does not need to be fixed: when it is "worth it" (i.e. when investing more resource reduces the expected behavioral cost more than it increases the expected neural cost), more resource may be invested.

*Equations (1) and (2)* specify the theory at the most general level. To derive testable predictions, we next propose specific formalizations of resource and of the two expected cost functions.

### Formalization of resource

As in our previous work (*Keshvari et al., 2012*; *Keshvari et al., 2013*; *Mazyar et al., 2012*; *van den Berg et al., 2012*; *van den Berg et al., 2014*), we quantify encoding precision as Fisher information, $J$. This measure provides a lower bound on the variance of any unbiased estimator (*Cover and Thomas, 2005*; *Ly et al., 2017*) and is a common tool in the study of theoretical limits on stimulus coding and discrimination (*Abbott and Dayan, 1999*). Moreover, we assume that there is item-to-item and trial-to-trial variation in precision (*Fougnie et al., 2012*; *van den Berg et al., 2012*; *van den Berg et al., 2014*; *Keshvari et al., 2013*; *van den Berg et al., 2017*). Following our previous work, we model this variability using a gamma distribution with a mean $\bar{J}$ and shape parameter $\tau \geq 0$ (larger $\tau$ means more variability); we denote this distribution by gamma $(J;\bar{J},\tau)$.

We specify resource vector $\mathbf{Q}$ as the vector with mean encoding precisions, $\bar{\mathbf{J}}$, such that the general theory specified by *Equations (1) and (2)* modifies to

$$\bar{C}_{\text{total}}\left(\bar{\mathbf{J}};\lambda,\tau\right) = \bar{C}_{\text{behavioral}}\left(\bar{\mathbf{J}};\tau\right) + \lambda \bar{C}_{\text{neural}}\left(\bar{\mathbf{J}};\tau\right) \tag{3}$$

and

$$\bar{\mathbf{J}}_{\text{optimal}} = \underset{\bar{\mathbf{J}}}{\text{argmin}}\, \bar{C}_{\text{total}}\left(\bar{\mathbf{J}};\lambda,\tau\right) \tag{4}$$

In this formulation, it is assumed that the brain has control over resource vector $\bar{\mathbf{J}}$, but not over the variability in how much resource is actually assigned to an item. It should be noted, however, that our choice to incorporate variability in $J$ is empirically motivated and not central to the theory: parameter $\tau$ mainly affects the kurtosis of the predicted estimation error distributions, not their variance or the way that the variance depends on set size (which is the focus of this paper). We will show that the theory also predicts set size effects when there is no variability in $J$.

### Formalization of expected neural cost

To formalize the neural cost function, we make two general assumptions. First, we assume that the expected neural cost induced by encoding a set of $N$ items is the sum of the expected neural cost

associated with each of the individual items. Second, we assume that each of these "local" neural costs has the same functional dependence on the amount of allocated resource: if two items are encoded with the same amount of resource, they induce equal amounts of neural cost. Combining these assumptions, the expected neural cost induced by encoding a set of $N$ items with resource $\bar{\mathbf{J}} = \{\bar{J}_1, \ldots, \bar{J}_N\}$ takes the form

$$\bar{C}_{\mathrm{neural}}(\bar{\mathbf{J}}; \tau) = \sum_{i=1}^{N} \bar{c}_{\mathrm{neural}}(\bar{J}_i; \tau),$$ (5)

where we introduced the convention to denote local costs (associated with a single item) with small $c$, to distinguish them from the global costs (associated with the entire set of encoded items), which we denote with capital $C$.

We denote by $c_{\mathrm{neural}}(J)$ the neural cost induced by investing an amount of resource $J$. The expected neural cost induced by encoding an item with resource $\bar{J}$ is obtained by integrating over $J$,

$$\bar{c}_{\mathrm{neural}}(\bar{J}; \tau) = \int c_{\mathrm{neural}}(J) \mathrm{Gamma}(J; \bar{J}, \tau) dJ,$$ (6)

The theory is agnostic about the exact nature of the cost function $c_{\mathrm{neural}}(J)$: it could include spiking and non-spiking components (*Lennie, 2003*), be associated with activity in both sensory and non-sensory areas, and include other types of cost that are linked to "mental effort" in general (*Shenhav et al., 2017*).

To motivate a specific form of this function, we consider the case that the neural cost is incurred by spiking activity. For many choices of spike variability, including the common one of Poisson-like variability (*Ma et al., 2006*), Fisher information $J$ of a stimulus encoded in a neural population is proportional to the trial-averaged neural spiking rate (*Paradiso, 1988*; *Seung and Sompolinsky, 1993*). If we further assume that each spike has a fixed cost, we find that the local neural cost induced by each item is proportional to $J$,

$$c_{\mathrm{neural}}(J; \alpha) = \alpha J,$$ (7)

where $\alpha$ is the amount of neural cost incurred by a unit increase in resource. Combining *Equations (5–7)* yields

$$\bar{C}_{\mathrm{neural}}(\bar{\mathbf{J}}; \alpha) = \alpha \sum_{i=1}^{N} \bar{J}_i.$$ (8)

Hence, the global expected neural cost is proportional to the total amount of invested resource and independent of the amount of variability in $J$. Although we use this linear expected neural cost function throughout the paper, we show in Appendix 1 that the key model prediction – a decrease of the optimal resource per item with set size – generalizes to a broad range of choices.

## Formalization of expected behavioral cost for local tasks

Before we specify the expected behavioral cost function, we introduce a distinction between two classes of tasks. First, we define a task as "local" if the observer's response depends on only one of the encoded items. Examples of local tasks are single-probe delayed-estimation (*Blake et al., 1997*; *Prinzmetal et al., 1998*; *Wilken and Ma, 2004*), single-probe change detection (*Todd and Marois, 2004*; *Luck and Vogel, 1997*), and single-probe change discrimination (*Klyszejko et al., 2014*). By contrast, when the task response depends on all memorized items, we define the task as "global." Examples of global tasks are whole-display change detection (*Luck and Vogel, 1997*; *Keshvari et al., 2013*), change localization (*van den Berg et al., 2012*), and delayed visual search (*Mazyar et al., 2012*). The theory that we developed up to this point – *Equations (1–8)* – applies to both global and local tasks. However, from here on, we develop our theory in the context of local tasks only; we will come back to global tasks at the end of the Results.

As in local tasks only one item gets probed, the expected behavioral cost across all items is a weighted average,

$$\bar{C}_{\text{behavioral}}(\bar{\mathbf{J}};\tau) = \sum_{i=1}^{N} p_i \bar{c}_{\text{behavioral},i}(\bar{J}_i;\tau), \tag{9}$$

where $p_i$ is the experimentally determined probing probability of the $i^{\text{th}}$ item and $\bar{c}_{\text{behavioral},i}(\bar{J}_i;\tau)$ is the local expected behavioral cost associated with reporting the $i^{\text{th}}$ item. We will refer to the product $p_i \bar{c}_{\text{behavioral},i}(\bar{J}_i;\tau)$ as the 'expected behavioral cost per item'. The only remaining step is to specify $\bar{c}_{\text{behavioral},i}(\bar{J}_i;\tau)$. This function is task-specific and we will specify it after we have described the task to which we apply the model.

## A resource-rational model for local tasks

Combining *Equations 3, 8, and 9* yields the following expected total cost function for local tasks:

$$\bar{C}_{\text{total}}(\bar{\mathbf{J}};\alpha,\lambda,\tau) = \sum_{i=1}^{N} p_i \bar{c}_{\text{behavioral}}(\bar{J}_i;\tau) + \alpha\lambda \sum_{i=1}^{N} \bar{J}_i. \tag{10}$$

As parameters $\alpha$ and $\lambda$ have interchangeable effects on the model predictions, we will fix $\alpha = 1$ and only treat $\lambda$ as a free parameter.

We recognize that the right-hand side of *Equation 10* is a sum of independent terms. Therefore, each element of $\bar{\mathbf{J}}_{\text{optimal}}$, *Equation 4*, can be computed independently of the other elements, by minimizing the expected total cost per item,

$$\bar{J}_{\text{optimal},i}(p_i;\lambda,\tau) = \underset{\bar{J}}{\text{argmin}}(p_i \bar{c}_{\text{behavioral}}(\bar{J};\tau) + \lambda\bar{J}). \tag{11}$$

This completes the specification of the general form of our resource-rational model for local tasks. Its free parameters are $\lambda$ and $\tau$.

## Set size effects result from cost minimization and are mediated by probing probability

To obtain an understanding of the model predictions, we analyze how $\bar{J}_{\text{optimal}}$ depends on probing probability and set size. We perform this analysis under two general assumptions about the local expected behavioral cost function: first, that it monotonically decreases with $\bar{J}$ (i.e. increasing resource reduces the expected behavioral cost) and, second, that it satisfies a law of diminishing returns (i.e. the reductions per unit increase of resource decrease with the total amount of already invested resource). It can be proven (see Appendix 1) that under these assumptions, the domain of probing probability $p_i$ consists of three potential regimes, each with a different optimal encoding strategy (*Figure 1A*). First, there might exist a regime $0 \leq p_i < p_0$ in which it is optimal to not encode an item, $\bar{J}_{\text{optimal}} = 0$. In this regime, the probing probability of an item is so low that investing any amount of resource can never reduce the local expected behavioral cost by more than it increases the expected neural cost. Second, there might exist a regime $p_0 \leq p_i < p_\infty$ in which it is optimal to encode an item with a finite amount of resource, $\bar{J}_{\text{optimal}} \in (0,\infty)$. In this regime, $\bar{J}_{\text{optimal}}$ increases as a function of $p_i$. Finally, there may be a regime $p_\infty \leq p_i \leq 1$ in which the optimal strategy is to encode the item with an infinite amount of resource, $\bar{J}_{\text{optimal}} = \infty$. This last regime will only exist in extreme cases, such as when there is no neural cost associated with encoding. The threshold $p_0$ depends on the importance of the neural cost, $\lambda$, and on the derivative of the local expected behavioral cost evaluated at $\bar{J} = 0$; specifically, $p_0 = \frac{\lambda}{|\bar{c}_{\text{behavioral}}'(0)|}$. The threshold $p_\infty$ depends on $\lambda$ and on the derivative of the local expected behavioral cost evaluated at $\bar{J} \rightarrow \infty$; specifically, $p_\infty = \frac{\lambda}{|\bar{c}_{\text{behavioral}}'(\infty)|}$. If $p_\infty > 1$, then the third regime does not exist, whereas if $p_0 > 1$, only the first regime exists.

We next turn to set size effects. An interesting property of the model is that $\bar{J}_{\text{optimal}}$ depends only on the probing probability, $p_i$, and on the model parameters – it does *not* explicitly depend on set size, $N$. Therefore, the only way in which the model can predict set size effects is through a coupling between $N$ and $p_i$. Such a coupling exists in most studies that use a local task. For example, in delayed-estimation tasks, each item is usually equally likely to be probed such that $p_i = 1/N$. For those experiments, the above partitioning of the domain of $p_i$ translates to a similar partitioning of the domain of $N$ (*Figure 1B*). Then, a set size $N_\infty \geq 0$ may exist below which it is optimal to encode

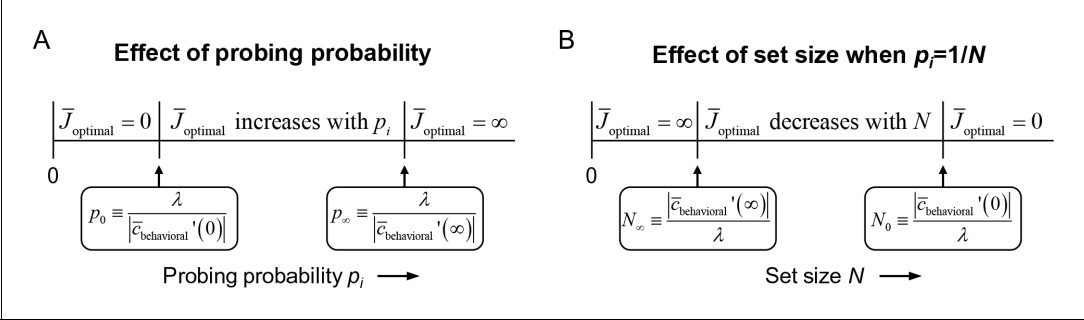

**Figure 1.** Effects of probing probability and set size on $\bar{J}_{\text{optimal}}$ in the resource-rational model for local tasks. (**A**) The model has three different optimal solutions depending on probing probability $p_i$: invest no resource when $p_i$ is smaller than some threshold value $p_0$, invest infinite resource when $p_i$ is larger than $p_\infty$, and invest a finite amount of resource when $p_0 < p_i < p_\infty$. The thresholds $p_0$ and $p_\infty$ depend on weight $\lambda$ (see *Equation (1)*) and on the derivative of the local expected behavioral cost function evaluated at 0 and $\infty$, respectively. If $p_0 > 1$, then only the first regime exists; if $p_0 < 1 < p_\infty$ then only the first two regimes exist. (**B**) If, in addition, $p_i = 1/N$, then the domain of $N$ partitions in a similar manner.
DOI: https://doi.org/10.7554/eLife.34963.003

items with infinite resource, a region $N_\infty \leq N < N_0$ in which it is optimal to encode items with a finite amount of resource, and a region $N > N_0$ in which it is optimal to not encode items at all.

## Results

### Model predictions for delayed-estimation tasks

To test the predictions of the model against empirical data, we apply it to the delayed-estimation task (*Wilken and Ma, 2004*; *Blake et al., 1997*; *Prinzmetal et al., 1998*), which is currently one of the most widely used paradigms in VWM research. In this task, the observer briefly holds a set of items in memory and then reports their estimate of a randomly probed target item (*Figure 2A*). Set size effects manifest as a widening of the estimation error distribution as the number of items is increased (*Figure 2B*), which suggests a decrease in the amount of resource per item (*Figure 2C*).

To apply our model to this task, we express the expected local behavioral cost as an expected value of the behavioral cost with respect to the error distribution,

$$\bar{c}_{\text{behavioral},i}(\bar{J}_i;\tau) = \int c_{\text{behavioral},i}(\varepsilon)p(\varepsilon;\bar{J}_i,\tau)d\varepsilon, \tag{12}$$

where the behavioral cost function $c_{\text{behavioral},i}(\varepsilon)$ maps an encoding error $\varepsilon$ to a cost and $p(\varepsilon;\bar{J}_i,\tau)$ is the predicted distribution of $\varepsilon$ for an item encoded with resource $\bar{J}_i$. We first specify $p(\varepsilon;\bar{J}_i,\tau)$ and then turn to $c_{\text{behavioral},i}(\varepsilon)$. As the task-relevant feature in delayed-estimation experiments is usually a circular variable (color or orientation), we make the common assumption that $\varepsilon$ follows a Von Mises distribution. We denote this distribution by $\text{VM}(\varepsilon;J)$, where $J$ is one-to-one related to the distribution's concentration parameter $\kappa$ (Appendix 1). The distribution of $\varepsilon$ for a stimulus encoded with resource $\bar{J}_i$ is found by integrating over $J$,

$$p(\varepsilon;\bar{J}_i,\tau) = \int \text{VM}(\varepsilon;J)\text{Gamma}(J;\bar{J}_i,\tau)dJ \tag{13}$$

Finally, we specify the behavioral cost function $c_{\text{behavioral},i}(\varepsilon)$ in *Equation 12*, which maps an estimation error $\varepsilon$ to a behavioral cost. As in most psychophysical experiments, human subjects tend to perform well on delayed-estimation tasks even when the reward is independent of their performance. This suggests that the behavioral cost function is strongly determined by internal incentives. A recent paper (*Sims, 2015*) has attempted to measure this mapping and proposed a two-parameter function. We will test that proposal later, but for the moment we assume a simpler, one-parameter power-law function, $c_{\text{behavioral},i}(\varepsilon;\beta) = |\varepsilon|^\beta$, where power $\beta$ is a free parameter.

To obtain an intuition for the predictions of this model, we plot in *Figure 2D* for a specific set of parameters the two expected costs per item and their sum, *Equation 11*, as a function of $\bar{J}$. The

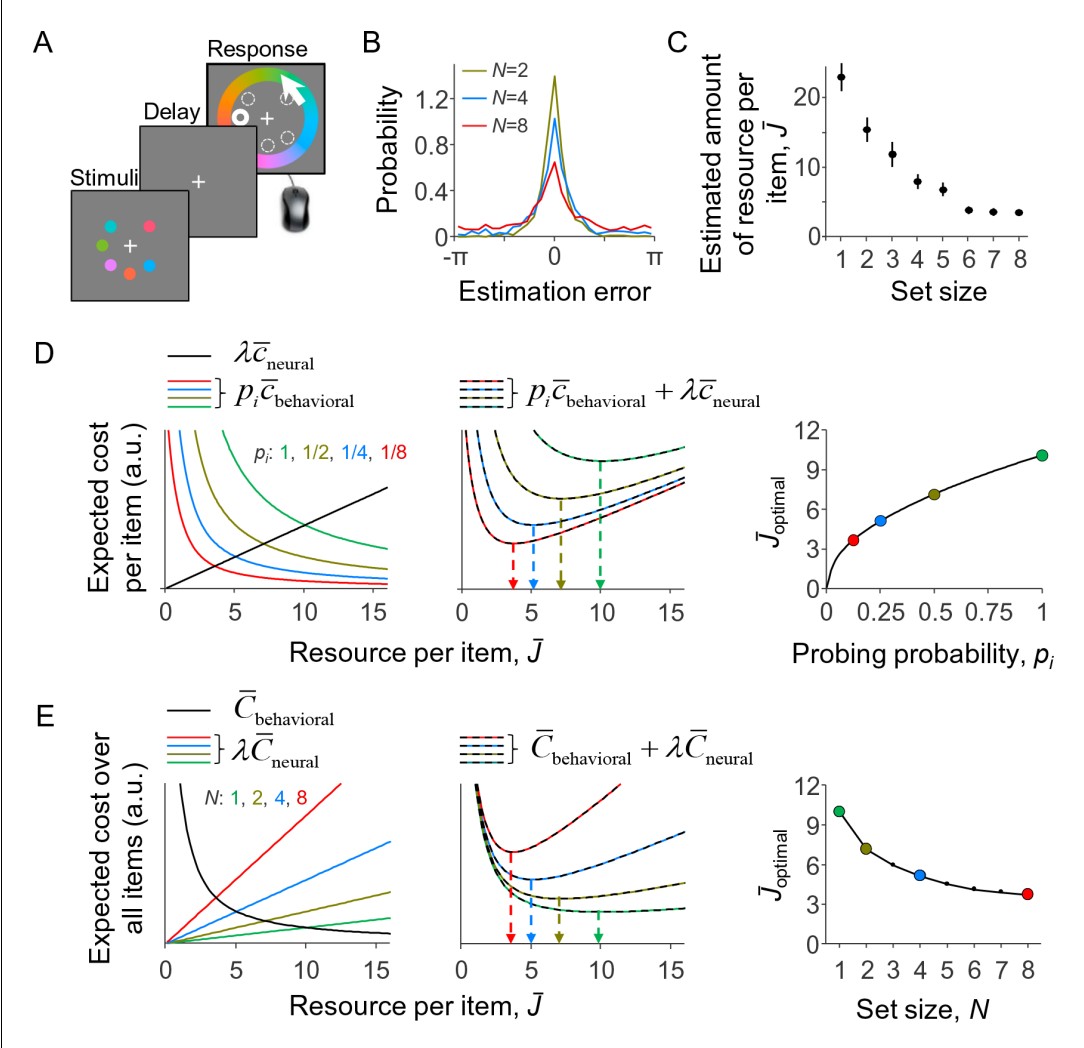

**Figure 2.** A resource-rational model for delayed-estimation tasks. (A) Example of a trial in a delayed-estimation experiment. The subject is briefly presented with a set of stimuli and, after a short delay, reports the value of the item at a randomly chosen location (here indicated with thick circle). (B) The distribution of estimation errors in delayed-estimation experiments typically widens with set size (data from Experiment E5 in **Table 1**). (C) This suggests that the amount of resource per encoded item decreases with set size. The estimated amount of resource per item was computed using the same non-parametric model as the one underlying **Figure 3C**. (D) Expected cost per item as a function of the amount of invested resource (model parameters: $\lambda$ = 0.01, $\beta$ = 2, $\tau$↓0). *Left*: The expected behavioral cost per item (colored curves) decreases with the amount of invested resource, while the expected neural cost per item increases (black line). *Center*: The sum of these two costs has a unique minimum, whose location (arrows) depends on probing probability $p_i$. *Right*: The optimal amount of resource per item increases with the probability that the item will be probed. (E) Expected cost across all items, when each item is probed with a probability $p_i$ = 1/N; the model parameters are the same as in **D** and the set sizes correspond with the values of $p_i$ in **D**. The predicted set size effect (right panel) is qualitatively similar to set size effects observed in empirical data (cf. panel C). (D) and (E) are alternative illustrations of the same optimization problem; the right panel of (E) could also be obtained by replotting the right panel of (D) as a function of N = 1/$p_i$.

DOI: https://doi.org/10.7554/eLife.34963.004

The following figure supplement is available for figure 2:

**Figure supplement 1.** Fits to the three delayed-estimation benchmark data sets that were excluded from the main analyses.
DOI: https://doi.org/10.7554/eLife.34963.006

expected behavioral cost per item depends on $p_i$ and decreases with $\bar{J}$ (colored curves in left panel), while the expected neural cost per item is independent of $p_i$ and increases (black line in left panel). The expected total cost per item has a unique minimum (middle panel). The value of $\bar{J}$ corresponding to this minimum, $\bar{J}_{optimal}$, increases with $p_i$ (**Figure 2D**, right). Hence, in this example, the optimal amount of resource per item is an increasing function of its probing probability.

We next consider the special case in which each item is equally likely to be probed, that is, $p_i = 1/N$. The values of $p_i$ in **Figure 2D** then correspond to set sizes 1, 2, 4, and 8. When replotting $\bar{J}_{\text{optimal}}$ as a function of $N$, we find a set size effect (**Figure 2E**, right panel) that is qualitatively similar to the empirical result in **Figure 2C**. An alternative way to understand this predicted set size effect is by considering how the three expected costs across all items, **Equation 3**, depend on $\bar{J}$. Substituting $p_i = 1/N$ in **Equation 9**, we find that the expected behavioral cost across all items is independent of set size (**Figure 2E**, left panel, black curve). Moreover, when all items are encoded with the same amount of resource (which is necessarily the optimal solution when $p_i$ is identical across items), the expected neural cost across all items equals $N\bar{J}$ and therefore scales linearly with set size (**Figure 2E**, left panel, colored lines). The sum of these terms has a unique minimum $\bar{J}_{\text{optimal}}$ (**Figure 2E**, center panel), which monotonically decreases with set size (**Figure 2E**, right panel). The costs plotted in **Figure 2E** can be considered as obtained by multiplying the corresponding costs in **Figure 2D** by $N$.

The model thus predicts set size effects in delayed-estimation tasks that are fully mediated by individual-item probing probability. The latter notion is consistent with empirical observations. *Palmer et al. (1993)* reported that "relevant set size" (where irrelevance means $p_i = 0$) acts virtually identically to actual set size. *Emrich et al. (2017)* independently varied probing probability and set size in their experiment, and found that the former was a better predictor of performance than the latter. Based on this, they hypothesized that set size effects are mediated by probing probability. The predictions of our model are qualitatively consistent with these findings.

## Model fits to data from delayed-estimation experiments with equal probing probabilities

To examine how well the model accounts for set size effects in empirical data, we fit it to data from six experiments that are part of a previously published benchmark set (E1-E6 in **Table 1**). We use a Bayesian optimization method (*Acerbi and Ma, 2017*) to estimate the maximum-likelihood parameter values, separately for each individual data set (see **Table 2** for a summary of these estimates). The model accounts well for the subject-level error distributions (**Figure 3A**) and the two statistics that summarize these distributions (**Figure 3B**). The original benchmark set (*van den Berg et al., 2014*) contained four more data sets, but three of those were published in papers that were later retracted and another one contains data at only two set sizes. Although we decided to leave those four datasets out of our main analyses, the model accounts well for them too (**Figure 2—figure supplement 1**).

We next compare the goodness of fit of the resource-rational model to that of a descriptive variant in which the amount of resource per item, $\bar{J}$, is assumed to be a power-law function of set size (all other aspects of the model are kept the same). This variant is identical to the VP-A model in our earlier work, which is one of the most accurate descriptive models currently available (*van den Berg et al., 2014*). Model comparison based on the Akaike Information Criterion (AIC) (*Akaike, 1974*) indicates that the data provide similar support for both models, with a small advantage for the

**Table 1.** Overview of experimental datasets.
Experiments E5 and E6 differed in the way that subjects provided their responses (E5: color wheel; E6: scroll).

| Exp. ID | Reference | Feature | Set size(s) | Probing probability | Number of subjects |
|---------|-----------|---------|-------------|---------------------|--------------------|
| E1 | *Wilken and Ma (2004)* | Color | 1, 2, 4, 8 | Equal | 15 |
| E2 | *Zhang and Luck (2008)* | Color | 1, 2, 3, 6 | Equal | 8 |
| E3 | *Bays et al. (2009)* | Color | 1, 2, 4, 6 | Equal | 12 |
| E4 | *van den Berg et al. (2012)* | Orientation | 1-8 | Equal | 6 |
| E5 | *van den Berg et al. (2012)* | Color | 1-8 | Equal | 13 |
| E6 | *van den Berg et al. (2012)* | Color | 1-8 | Equal | 13 |
| E7 | *Bays et al. (2009)* | Orientation | 2,4,8 | Unequal | 7 |
| E8 | *Emrich et al. (2017)* | Color | 4 | Unequal | 20 |
| E9 | *Emrich et al. (2017)* | Color | 6 | Unequal | 20 |

DOI: https://doi.org/10.7554/eLife.34963.005

van den Berg and Ma. eLife 2018;7:e34963. DOI: https://doi.org/10.7554/eLife.34963

**Table 2.** Subject-averaged parameter estimates of the resource-rational model fitted to data from nine previously published experiments.
See *Table 1* for details about the experiments.

| Experiment | $\beta$ | $\lambda$ | $\tau$ |
|---|---|---|---|
| E1 | $1.87 \pm 0.29$ | $(4.8 \pm 1.2) \cdot 10^{-2}$ | $17.9 \pm 2.5$ |
| E2 | $(1.33 \pm 0.30) \cdot 10^{-2}$ | $(4.27 \pm 0.83) \cdot 10^{-4}$ | $14.8 \pm 1.1$ |
| E3 | $0.138 \pm 0.042$ | $(2.78 \pm 0.87) \cdot 10^{-3}$ | $19.1 \pm 2.6$ |
| E4 | $0.106 \pm 0.052$ | $(3.2 \pm 1.4) \cdot 10^{-3}$ | $8.2 \pm 1.8$ |
| E5 | $0.356 \pm 0.085$ | $(5.8 \pm 1.1) \cdot 10^{-3}$ | $18.1 \pm 2.8$ |
| E6 | $0.61 \pm 0.15$ | $(8.8 \pm 1.5) \cdot 10^{-3}$ | $7.4 \pm 1.3$ |
| E7 | $1.19 \pm 0.51$ | $(9.5 \pm 6.6) \cdot 10^{-2}$ | $5.7 \pm 1.5$ |
| E8 | $0.58 \pm 0.19$ | $(1.58 \pm 0.66) \cdot 10^{-2}$ | $27.0 \pm 3.7$ |
| E9 | $0.93 \pm 0.25$ | $(3.0 \pm 1.0) \cdot 10^{-2}$ | $23.7 \pm 2.3$ |

DOI: https://doi.org/10.7554/eLife.34963.008

resource-rational model ($\Delta$AIC = 5.27 $\pm$ 0.70; throughout the paper, X $\pm$ Y indicates mean $\pm$s.e.m. across subjects). Hence, the resource-rational model provides a principled explanation of set size effects without sacrificing quality of fit compared to one of the best available descriptive models of VWM. We find that the resource-rational model also fits better than a model in which the total amount of resource is fixed and divided equally across items ($\Delta$AIC = 13.9 $\pm$ 1.4).

So far, we have assumed that there is random variability in the actual amount of resource assigned to an item. Next, we test an equal-precision variant of the resource-rational model, by fixing parameter $\tau$ to a very small value ($10^{-3}$). Consistent with the results obtained with the variable-precision model, we find that the rational model has a substantial AIC advantage over a fixed-resource model ($\Delta$AIC = 43.0 $\pm$ 6.8) and is on equal footing with the power-law model ($\Delta$AIC = 2.0 $\pm$ 1.7 in favor of the power-law model). However, all three equal-precision models (fixed resource, power law, rational) are outperformed by their variable-precision equivalents by over 100 AIC points. Therefore, we will only consider variable-precision models in the remainder of the paper.

To get an indication of the absolute goodness of fit of the resource-rational model, we next examine how much room for improvement there is in the fits. We do this by fitting a non-parametric model variant in which resource $\bar{J}$ is a free parameter at each set size, while keeping all other aspects of the model the same. We find a marginal AIC difference, suggesting that the fits of the rational model cannot be improved much further without overfitting the data ($\Delta$AIC = 3.49 $\pm$ 0.93, in favor of the non-parametric model). An examination of the fitted parameter values corroborates this finding: the estimated resource values in the non-parametric model closely match the optimal values in the rational model (*Figure 3C*).

So far, we have assumed that behavioral cost is a power-law function of the absolute estimation error, $c_{behavioral}(\varepsilon) = |\varepsilon|^{\beta}$. To evaluate the necessity of a free parameter in this function, we also test three parameter-free choices: $|\varepsilon|$, $\varepsilon^2$, and $-\cos(\varepsilon)$. Model comparison favors the original model with AIC differences of 14.0 $\pm$ 2.8, 24.4 $\pm$ 4.1, and 19.5 $\pm$ 3.5, respectively. While there may be other parameter-free functions that give better fits, we expect that a free parameter is unavoidable here, as the error-to-cost mapping may differ across experiments (because of differences in external incentives) and also across subjects within an experiment (because of differences in intrinsic motivation). Finally, we also test a two-parameter function that was proposed recently (Equation (5) in *Sims [2015]*). The main difference with our original choice is that this alternative function allows for saturation effects in the error-to-cost mapping. However, this extra flexibility does not increase the goodness of fit sufficiently to justify the additional parameter, as the original model outperforms this variant with an AIC difference of 5.3 $\pm$ 1.8.

Finally, we use five-fold cross validation to verify the AIC-based results reported in this section. We find that they are all consistent (*Table 3*).

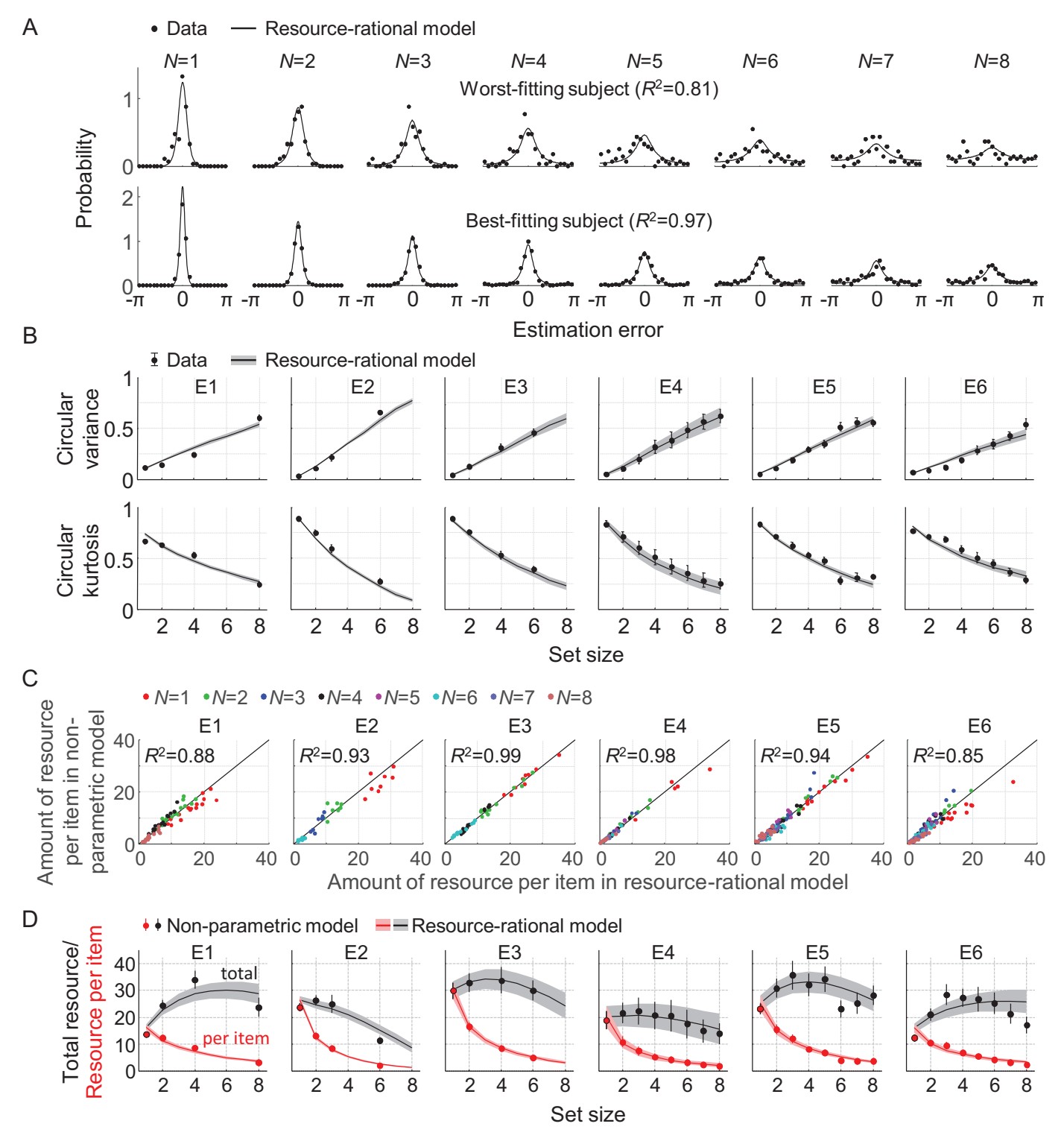

**Figure 3.** Model fits to data from six delayed-estimation experiments with equal probing probabilities. (A) Maximum-likelihood fits to raw data of the worst-fitting and best-fitting subjects (subjects S10 in E6 and S4 in E4, respectively). Goodness of fit was measured as $R^2$, computed for each subject by concatenating histograms across set sizes. (B) Subject-averaged circular variance and kurtosis of the estimation error, as a function of set size and split by experiment. The maximum-likelihood fits of the model account well for the trends in these statistics. (C) Estimated amounts of resource per item in the resource-rational model scattered against the estimates in the non-parametric model. Each dot represents estimates from a single subject. (D)
*Figure 3 continued on next page*

*Figure 3 continued*

Estimated amount of resource per item (red) and total resource (black) plotted against set size. Here and in subsequent figures, error bars and shaded areas represent 1 s.e.m. of the mean across subjects.

DOI: https://doi.org/10.7554/eLife.34963.007

## Non-monotonic relation between total resource and set size

One quantitative feature that sets the resource-rational theory apart from previous theories is its predicted relation between set size and the total amount of invested resource, $\bar{J}_{\text{total}} = \sum_{i=1}^{N} \bar{J}_i$. This quantity is by definition constant in fixed-resource models, and in power-law models it varies monotonically with set size. By contrast, we find that in the fits to several of the experiments, $\bar{J}_{\text{total}}$ varies *non-monotonically* with set size (*Figure 3D*, gray curves). To examine whether there is evidence for non-monotonic trends in the subject data, we next compute an "empirical" estimate $\bar{J}_{\text{total}} = \sum_{i=1}^{N} \hat{\bar{J}}_i$, where $\hat{\bar{J}}_i$ are the best-fitting resource estimates in the non-parametric model. We find that these estimates show evidence of similar non-monotonic relations in some of the experiments (*Figure 3D*, black circles). To quantify this evidence, we perform Bayesian paired t-tests in which we compare the estimates of $\bar{J}_{\text{total}}$ at set size 3 with the estimates at set sizes 1 and 6 in the experiments that included these three set sizes (E2 and E4-E6). These tests reveal strong evidence that the total amount of resource is higher at set size 3 than at set sizes 1 ($BF_{+0} = 1.05 \cdot 10^7$) and 6 ($BF_{+0} = 4.02 \cdot 10^2$). We next compute for each subject the set size at which $\bar{J}_{\text{total}}$ is largest, which we denote by $N_{\text{peak}}$, and find a subject-averaged value of $3.52 \pm 0.18$. Altogether, these findings suggest that the total amount of resource that subjects spend on item encoding varies non-monotonically with set size, which is consistent with predictions from the resource-rational model, but not with any of the previously proposed models. To the best of our knowledge, evidence for a possible non-monotonicity in the relation between set size and total encoding resource has not been reported before.

## Predicted effects of probing probability

As we noted before, the model predictions do not explicitly depend on set size, *N*. Yet, we found that the model accounts well for set size effects in the experiments that we considered so far (E1-E6). This happens because in all those experiments, *N* was directly coupled with probing probability $p_i$, through $p_i = 1/N$. This coupling makes it impossible to determine whether changes in subjects' encoding precision are the result of changes in *N* or changes in $p_i$. Therefore, we will next consider experiments in which individual probing probabilities and set size were varied independently of each other (E7-E9 in *Table 1*). According to our model, the effects of *N* that we found in E1-E6 were really effects of $p_i$. Therefore, we should be able to make predictions about effects of $p_i$ in E7-E9 by

**Table 3.** Comparing two metrics for model comparison: AIC and five-fold cross-validated log likelihood.

Each comparison is between the main version of the resource-rational model, *Equation (11)*, and the model listed in the first column of the table. Negative AIC differences and positive cross-validated log likelihood differences indicate an advantage of the resource-rational model over the alternative model. In all comparisons, these differences have opposite signs, which means that the AIC-based results are consistent with the cross-validation results.

| Model with which the main model is compared | AIC difference | Cross-validation log likelihood difference |
| --- | --- | --- |
| Descriptive power-law model | −5.27±0.70 | 2.59±0.39 |
| Descriptive fixed-resource model | −13.9±1.4 | 8.4±1.0 |
| Descriptive unconstrained model | 3.49±0.93 | −1.26±0.49 |
| Rational model variant: equal precision | −110±10 | 56±4.7 |
| Rational model variant: $c_{\text{behavioral}}=|\varepsilon|$ | −14±2.8 | 7.1±1.4 |
| Rational model variant: $c_{\text{behavioral}}=\varepsilon^2$ | −24.4±4.1 | 12.2±2.0 |
| Rational model variant: $c_{\text{behavioral}}=-\cos(\varepsilon)$ | −19.5±3.5 | 9.8±1.8 |
| Rational model variant: $c_{\text{behavioral}}$ as in Sims (2015) | −5.3±1.8 | 4.7±0.74 |

DOI: https://doi.org/10.7554/eLife.34963.009

recasting the effects of $N$ in E1-E6 as effects of $p_i = 1/N$. Given that the amount of resource per item in E1-E6 decreases with $N$, a first prediction is that it should increase as a function of $p_i$ in E7-E9. A second and particularly interesting prediction is that the estimated total amount of invested resource should vary non-monotonically with $p_i$ and peak at a value $p_{peak}$ that is close to $1/N_{peak}$ found in E1-E6 (see previous section). Based on the values of $N_{peak}$ in experiments E1-E6, we find a prediction $p_{peak} = 0.358 \pm 0.026$.

## Model fits to data from delayed-estimation experiments with unequal probing probabilities

To test the predictions presented in the previous section and, more generally, to evaluate how well our model accounts for effects of $p_i$ on encoding precision, we fit it to data from three experiments in which probing probability was varied independently of set size (E7-E9 in *Table 1*).

In the first of these experiments (E7), seven subjects performed a delayed-estimation task at set sizes 2, 4, and 8. On each trial, one of the items – indicated with a cue – was three times more likely to be probed than any of the other items. Hence, the probing probabilities for the cued and uncued items were 3/4 and 1/4 at $N = 2$, respectively, 1/2 and 1/6 at $N = 4$, and 3/10 and 1/10 at $N = 8$. The subject data show a clear effect of $p_i$: the higher the probing probability of an item, the more precise the subject responses (*Figure 4A*, top row, black circles). We find that the resource-rational model, *Equation (11)*, accounts well for this effect (*Figure 4A*, top row, curves) and does so by increasing the amount of resource as a function of probing probability $p_i$ (*Figure 4B*, left panel, red curves).

In the other two experiments (E8 and E9), the number of cued items and cue validity were varied between conditions, while set size was kept constant at 4 or 6. For example, in one of the conditions of E8, three of the four items were cued with 100% validity, such that $p_i$ was 1/3 for each cued item and 0 for the uncued item; in another condition of the same experiment, two of the four items were cued with 66.7% validity, meaning that $p_i$ was 1/3 for each cued item and 1/6 for each uncued item. The unique values of $p_i$ across all conditions were {0, 1/6, 2/9, 1/4, 1/3, 1/2, 1} in E8 and {0, 1/12, 1/10, 2/15, 1/6, 1/3, 1/2, and 1} in E9. As in E7, responses become more precise with increasing $p_i$ and the model accounts well for this (*Figure 4A*), again by increasing the amount of resource assigned to an item with $p_i$ (*Figure 4B*).

We next examine how our model compares to the models proposed in the papers that originally published these three data sets. In contrast to our model, both *Bays (2014)* and *Emrich et al. (2017)* proposed that the total amount of invested resource is fixed. However, while Bays proposed that the distribution of this resource is in accordance with minimization of a behavioral cost function (as in our model), Emrich et al. postulated that the resource is distributed in proportion to each item's probing probability. Hence, while our model optimizes both the amount of invested resource and its distribution, Bays' model only optimizes the distribution, and Emrich et al.'s model does not explicitly optimize anything. To examine how the three proposals compare in terms of how well they account for the data, we fit two variants of our model that encapsulate the main assumptions of these two earlier proposals. In the first variant, we compute $\bar{\mathbf{J}}_{optimal}$ as $\underset{bf\bar{J}}{\operatorname{argmin}} \left[ \sum_{i=1}^{N} p_i \bar{c}_{behavioral}(\bar{J}_i; \beta, \tau) \right]$

under the constraint $\sum_{i=1}^{N} \bar{J}_i = \bar{J}_{total}$, which is consistent with Bays' proposal. Hence, in this variant, the neural cost function is removed and parameter $\lambda$ is replaced by a parameter $\bar{J}_{total}$ – otherwise, all aspects of the model are the same as in our main model. In the variant that we use to test Emrich et al.'s proposal, we compute $\bar{J}_i$ for each item as $p_i \bar{J}_{total}$, where $p_i$ is the probing probability and $\bar{J}_{total}$ is again a free parameter that represents the total amount of resource. Fitting the models to the data from all 47 subjects in E7-E9, we find a substantial advantage of our model over the proposal by Emrich et al., with an AIC difference of $18.0 \pm 3.9$. However, our model cannot reliably be distinguished from the proposal by Bays: either model is preferred in about half of the subjects (our model: 27; Bays: 20) and the subject-averaged AIC difference is negligible ($1.8 \pm 2.5$ in favor of our model). Hence, the model comparison suggests quite convincingly that subjects distribute their resource near-optimally across items with unequal probing probabilities, but it is inconclusive regarding the question of whether the total amount of invested resource is fixed or optimized.

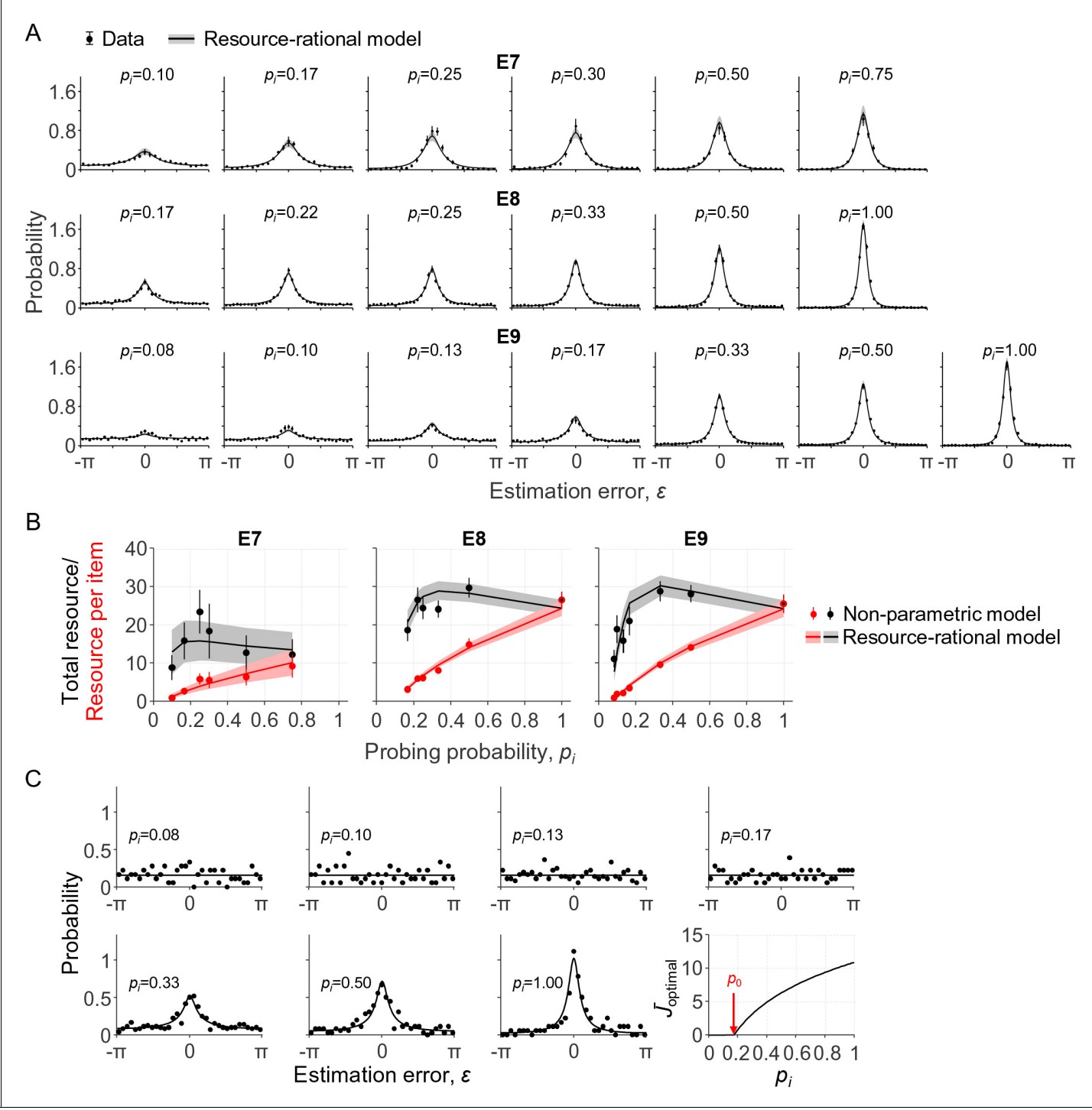

**Figure 4.** Model fits to data from three delayed-estimation experiments with unequal probing probabilities. (**A**) Fits of the resource-rational model (curves) to the data (black circles) of experiments E7-E9. (**B**) Estimated amount of resource per item as a function of probing probability (red) and the corresponding estimated total amount of resource that the subject would spend on encoding a display filled with items with equal probing probabilities (black). (**C**) Error histograms and a plot of $\bar{J}_{optimal}$ as a function of $p_i$ for a single subject (S4 in E9). The estimated value of $p_0$ was 0.18 for this subject, which was larger than the smallest probing probability in the experiment. The error histograms for items with the four lowest probing probabilities appear to be uniform for this subject, which is indicative of guessing (p>0.23 in Kolgomorov-Smirnov tests for uniformity on these four distributions).

DOI: https://doi.org/10.7554/eLife.34963.010

As an alternative way to address the question of whether the total amount of resource is fixed, we again fit a non-parametric model to obtain "empirical" estimates of the total amount of invested resource. To this end, we define $\hat{\bar{J}}_{\text{total}} = \hat{\bar{J}}_i / p_i$, where $\hat{\bar{J}}_i$ are the best-fitting values in a non-parametric model, such that $\hat{\bar{J}}_{\text{total}}$ represents the estimated total amount of resource that a subject would invest to encode a display filled with items that all have probing probability $p_i$. We find that these estimates show signs of a non-monotonicity as a function of $p_i$ (*Figure 4B*, black points), which are captured reasonably well by the resource-rational model (*Figure 4B*, black curves). Averaged across all subjects in E7-E9, the value of $p_i$ at which $\hat{\bar{J}}_{\text{total}}$ is largest is $0.384 \pm 0.037$, which is close to the predicted value of $0.358 \pm 0.026$ (see previous section). Indeed, a Bayesian independent-samples t-test supports the null hypothesis that there is no difference ($BF_{01} = 4.27$). Hence, while the model comparison results in the previous paragraph were inconclusive regarding the question of whether the total amount of invested resource is fixed or optimized, the present analysis provides evidence against fixed-resource models and confirms a prediction made by our own model.

In summary, the results in this section show that effects of probing probability in E7-E9 are well accounted for by the same model as we used to explain effects of set size in E1-E6. Regardless of whether total resource is fixed or optimized, this finding provides further support for the suggestion that set size effects are mediated by probing probability (*Emrich et al., 2017*) or, more generally, by item relevance (*Palmer et al., 1993*).

### Is it ever optimal to not encode an item?

There is an ongoing debate about the question of whether a task-relevant item is sometimes completely left out of working memory (*Adam et al., 2017*; *Luck and Vogel, 2013*; *Ma et al., 2014*; *Rouder et al., 2008*). Specifically, slot models predict that this happens when set size exceeds the number of slots (*Zhang and Luck, 2008*). In resource models, the possibility of complete forgetting has so far been an added ingredient separate from the core of the model (*van den Berg et al., 2014*). Our normative theory allows for a reinterpretation of this question: are there situations in which it is optimal to assign zero resource to the encoding of an item? We already established that this could happen in delayed-estimation tasks: whenever the probing probability is lower than a threshold value $p_0 = \frac{\lambda}{|\dot{c}_{\text{behavioral}}(0)|}$, the optimal amount of resource to invest on encoding the item is zero (see Theory). But what values does $p_0$ take in practice? Considering the expected behavioral cost function of a fixed-precision model (a variable-precision model with $\tau \downarrow 0$), we can prove that $p_0 = 0$, that is, it is never optimal to invest no resource (Appendix 1). For the expected behavioral cost function of the variable-precision model, however, simulations indicate that $p_0$ can be greater than 0 (we were not able to derive this result analytically). We next examine whether this ever happens under parameter values that are representative for human subjects. Using the maximum-likelihood parameters obtained from the data in E7-E9, we estimate that $p_0$ (expressed as a percentage) equals $8.86 \pm 0.54\%$. Moreover, we find that for 8 of the 47 subjects, $p_0$ is larger than the lowest probing probability in the experiment, which suggests that these subjects sometimes entirely ignored one or more of the items. For these subjects, the error distributions on items with $p_i < p_0$ look uniform (see *Figure 4C* for an example) and Kolmogorov-Smirnov tests for uniformity did not reject the null hypothesis in any of these cases ($p > 0.05$ in all tests).

These results suggest that there might be a principled reason why people sometimes leave task-relevant items out of visual working memory in delayed-estimation experiments. However, our model cannot explain all previously reported evidence for this. In particular, when probing probabilities are equal for all items, the model makes an "all or none" prediction: all items are encoded when $p_i > p_0$ and none are encoded otherwise. Hence, the model cannot explain why subjects in tasks with equal probing probabilities sometimes seem to encode a subset of task-relevant items. For example, a recent study reported that in a whole-report delayed-estimation experiment ($p_i = 1$ for all items), subjects encoded about half of the six presented items on each trial (*Adam et al., 2017*). Unless additional assumptions are made, our model cannot account for this finding.

### Predictions for a global task: whole-display change detection

The results so far show that the resource-rational model accounts well for data in a variety of delayed-estimation experiments. To examine how its predictions generalize to other tasks, we next consider a change detection task, which is another widely used paradigm in research on VWM. In

this task, the observer is sequentially presented with two sets of items and reports if any one of them changed (*Figure 5A*). In the variant that we consider here, a change is present on exactly half of the trials and is equally likely to occur in any of the items. We construct a model for this task by combining *Equations 3, 4, and 8* with an expected behavioral cost function based on the Bayesian decision rule for this task (see Appendix 1), which yields

$$\bar{\mathbf{J}}_{\text{optimal}} = \underset{\bar{\mathbf{J}}}{\arg\min} \left[ p\left(\text{error}|\bar{\mathbf{J}}\right) + \lambda \sum_{i=1}^{N} \bar{J}_i \right], \tag{14}$$

where $p\left(\text{error}|\bar{\mathbf{J}}\right)$ is the expected behavioral cost function, which in this case specifies the probability of an error response when a set of items is encoded with resource $\bar{\mathbf{J}}$.

In contrast to local tasks, the expected total cost in global tasks cannot be written as a sum of expected costs per item, because the expected behavioral cost – such as $p\left(\text{error}|\bar{\mathbf{J}}\right)$ in *Equation (14)* – can only be computed globally, not per item. Consequently, the elements of $\bar{\mathbf{J}}_{\text{optimal}}$ in global tasks

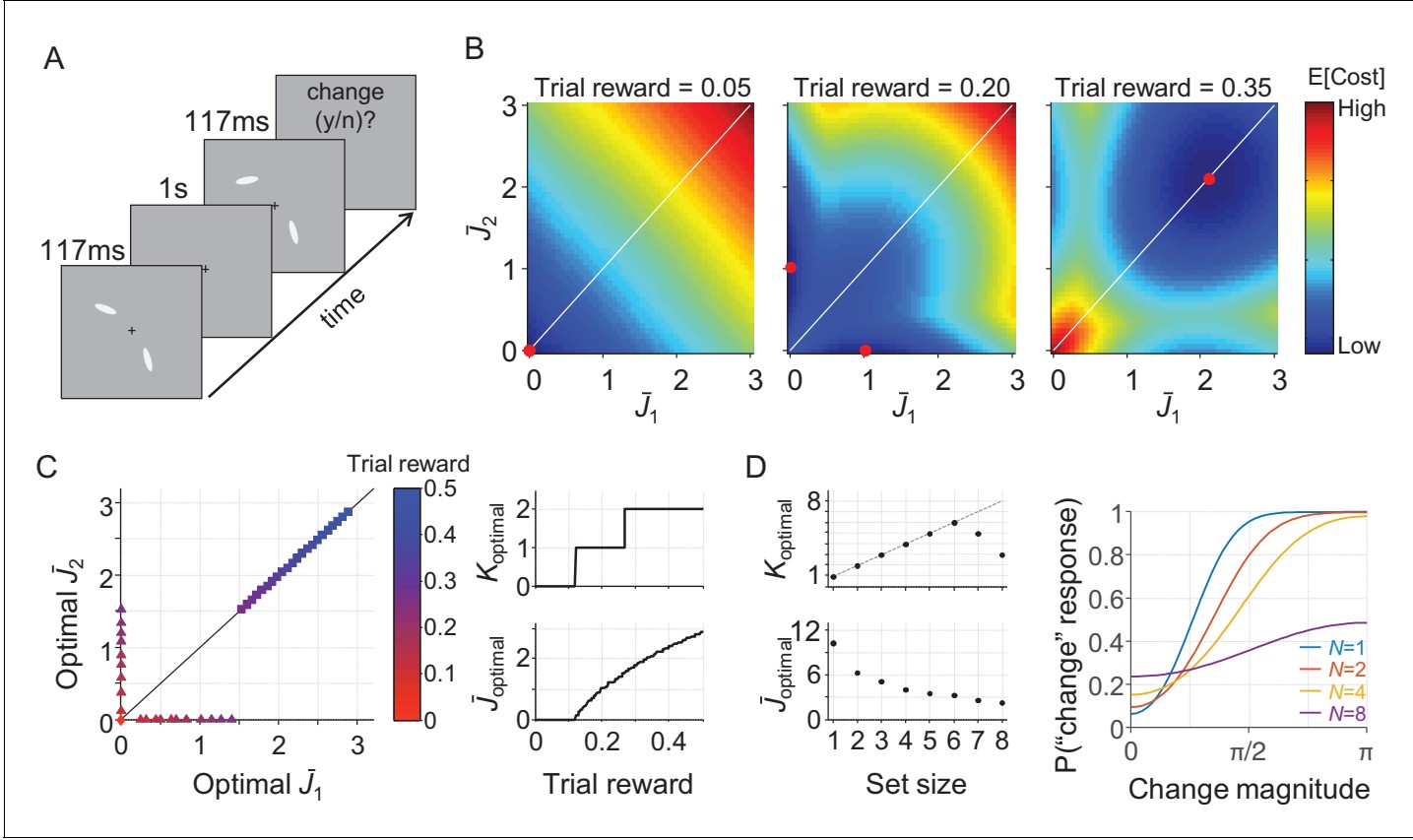

**Figure 5.** A resource-rational model for change detection tasks. (**A**) Example of a trial in a change detection task with a set size of 2. The subject is sequentially presented with two sets of stimuli and reports whether there was a change at any of the item locations. (**B**) Simulated expected total cost in the resource-rational cost function applied to a task with a set size of 2 and a reward of 0.05 (left), 0.20 (center), or 0.35 (right) units per correct trial. The red dot indicates the location of minimum cost, that is the resource-optimal combination of $\bar{J}_1$ and $\bar{J}_2$ (note that the expected cost function in the central panel has a minimum at two distinct locations). When reward is low (left), the optimal strategy is to encode neither of the two stimuli. When reward is high (right), the optimal strategy is to encode both stimuli with equal amounts of resource. For intermediate reward (center), the optimal strategy is to encode one of the two items, but not the other one. (**C**) Model predictions as a function of trial rewards at $N = 2$. Left: The amount of resource assigned to the two items for a range of reward values. Right: the corresponding optimal number of encoded items (top) and optimal amount of resource per encoded item (bottom) as a function of reward. (**D**) Model predictions as a function of set size (trial reward = 1.5). The model predicts set size effects in both the number of encoded items (left, top) and the amount of resource with which these items are encoded (left, bottom). Moreover, the model produces response data (right) that are qualitatively similar to human data (see, for example, *Figure 2C* in *Keshvari et al., 2013*). The parameter values used in all simulations were $\lambda$ = 0.01 and $\tau{\downarrow}0$.
DOI: https://doi.org/10.7554/eLife.34963.011

cannot be computed separately for each item. This makes resource optimization computationally much more demanding, because it requires solving an $N$-dimensional minimization problem instead of $N$ one-dimensional problems.

We perform a simulation at $N = 2$ (which is still tractable) to get an intuition of the predictions that follow from *Equation (14)*. For practical convenience, we assume in this simulation that there is no variability in precision, $\tau\downarrow 0$, such that $\lambda$ is the only model parameter. The results (*Figure 5B*) show that the cost-minimizing strategy is to encode neither of the items when the amount of reward per correct trial is very low (left panel) and encode them both when reward is high (right panel). However, interestingly, there is also an intermediate regime in which the optimal strategy is to encode one of the two items, but not the other one (*Figure 5B*, central panel). Hence, just as in the delayed-estimation task, there are conditions in which it is optimal to encode only a subset of items. An important difference, however, is that in the delayed-estimation task this only happens when items have unequal probing probabilities, while in this change detection task it even happens when all items are equally likely to change.

Simulations at larger set sizes quickly become computationally intractable, because of the reason mentioned above. However, the results at $N = 2$ suggest that if two items are encoded, the optimal solution is to encode them with the same amount of resource (*Figure 5C*). Therefore, we conjecture that all non-zero values in $\bar{\mathbf{J}}_{\text{optimal}}$ are identical, which would mean that the entire vector can be summarized by two values: the number of encoded items, which we denote by $K_{\text{optimal}}$, and the amount of resource assigned to each encoded item, which we denote by $\bar{J}_{\text{optimal}}$. Using this conjecture (which we have not yet been able to prove), we are able to efficiently compute predictions at an arbitrary set size. Simulation results show that the model then predicts that both $K_{\text{optimal}}$ and $\bar{J}_{\text{optimal}}$ depend on set size (*Figure 5D*, left) and produces response data that are qualitatively similar to human data (*Figure 5D*, right).

## Discussion

### Summary

Descriptive models of visual working memory (VWM) have evolved to a point where there is little room for improvement in how well they account for experimental data. Nevertheless, the basic finding that VWM precision depends on set size still lacks a principled explanation. Here, we examined a normative proposal in which expected task performance is traded off against the cost of spending neural resource on encoding. We used this principle to construct a resource-rational model for "local" VWM tasks and found that set size effects in this model are fully mediated by the probing probabilities of the individual items; this is consistent with suggestions from earlier empirical work (*Emrich et al., 2017*; *Palmer et al., 1993*). From the perspective of our model, the interpretation is that as more items are added to a task, the relevance of each individual item decreases, which makes it less cost-efficient to spend resource on its encoding. We also found that in this model it is sometimes optimal to encode only a subset of task-relevant items, which implies that resource rationality could serve as a principled bridge between resource and slot-based models of VWM. We tested the model on data from nine previous delayed-estimation experiments and found that it accounts well for effects of both set size and probing probability, despite having relatively few parameters. Moreover, it accounts for a non-monotonicity that appears to exist between set size and the total amount of resource that subjects invest in item encoding. The broader implication of our findings is that VWM limitations – and cognitive limitations in general – may be driven by a mechanism that minimizes a cost, instead of by a fixed constraint on available encoding resource.

### Limitations

Our theory makes a number of assumptions that need further investigation. First, we have assumed that the expected behavioral cost decreases indefinitely with the amount of invested resource, such that in the limit of infinite resource there is no encoding error and no behavioral cost. However, encoding precision in VWM is fundamentally limited by the precision of the sensory input, which is itself limited by irreducible sources of neural noise – such as Johnson noise and Poisson shot noise (*Faisal et al., 2008*; *Smith, 2015*) – and suboptimalities in early sensory processing (*Beck et al., 2012*). One way to incorporate this limitation is by assuming that there is a resource value $\bar{J}_{\text{input}}$

beyond which the expected behavioral cost no longer decreases as a function of $\bar{J}$. In this variant, $\bar{J}_{\text{input}}$ represents the quality of the input and $\bar{J}_{\text{optimal}}$ will never exceed this value, because any additional resource would increase the expected neural cost without decreasing the expected behavioral cost.

Moreover, our theory assumes that there is no upper limit on the total amount of resource available for encoding: cost is the only factor that matters. However, as the brain is a finite entity, the total amount of resource must obviously have an upper limit. This limit can be incorporated by optimizing $\mathbf{J}_{\text{optimal}}$ under the constraint $\sum_{i=1}^{N} \bar{J}_{\text{optimal},i} \leq \bar{J}_{max}$, where $\bar{J}_{\text{max}}$ represents the maximum amount of resource that can be invested. While an upper limit certainly exists, it may be much higher than the average amount of resource needed to encode information with the same fidelity as the sensory input. If that is the case, then $\bar{J}_{\text{input}}$ would be the constraining factor and $\bar{J}_{\text{max}}$ would have no effect.

Similarly, our theory assumes that there is no lower limit on the amount of resource available for encoding. However, there is evidence that task-irrelevant stimuli are sometimes automatically encoded (*Yi et al., 2004*; *Shin and Ma, 2016*), perhaps because in natural environments few stimuli are ever completely irrelevant. This would mean that there is a lower limit to the amount of resource spent on encoding. In contradiction to the predictions of our model, such a lower limit would prevent subjects from sometimes encoding nothing at all. For local tasks, such a lower limit can be incorporated by assuming that probing probability $p_i$ is never zero.

We have fitted our model only to data from delayed-estimation experiments. However, it applies without modification to other local tasks, such as single-probe change detection (*Luck and Vogel, 1997*; *Todd and Marois, 2004*) and single-probe change discrimination (*Klyszejko et al., 2014*). Further work is needed to examine how well the model accounts for empirical data of such tasks. Moreover, it should further examine how the theory generalizes to global tasks. One such task could be whole-report change detection; we presented simulation results for this task but the theory remains to be further worked out and fitted to the data.

A final limitation is that our theory assumes that items are uniformly distributed and uncorrelated. Although this is correct for most experimental settings, items in more naturalistic settings are often correlated and can take non-uniform distributions. In such environments, the expected total cost can probably be further minimized by taking into account statistical regularities (*Orhan et al., 2014*). Moreover, recent work has suggested that even when items are uncorrelated and uniformly distributed, the expected estimation error can sometimes be reduced by using a "chunking" strategy, that is, encoding similar items as one (*Nassar et al., 2018*). However, as Nassar et al. assumed a fixed total resource and did not take neural encoding cost into account in their optimization, it remains to be seen whether chunking is also optimal in the kind of model that we proposed. We speculate that this is likely to be the case, because encoding multiple items as one will reduce the expected neural cost (fewer items to encode), while the increase in expected behavioral cost will be negligible if the items are very similar. Hence, it seems worthwhile to examine models that combine resource rationality with chunking.

## Variability in resource assignment

Throughout the paper, we have assumed that there is variability in resource assignment. Part of this variability is possibly a result of stochastic factors, but part of it may also be systematic – for example, particular colors and orientations may be encoded with higher precision than others (*Bae et al., 2014*; *Girshick et al., 2011*). Whereas the systematic component could have a rational basis (e.g. higher precision for colors and orientations that occur more frequently in natural scenes [*Ganguli and Simoncelli, 2010*; *Wei and Stocker, 2015*]), this is unlikely to be true for the random component. Indeed, when we jointly optimize $\bar{J}$ and $\tau$ in *Equation 11*, we find estimates of $\tau$ that consistently approach 0, meaning that any variability in encoding precision is suboptimal under our proposed cost function. One way to reconcile this apparent suboptimality with the otherwise normative theory is to postulate that maintaining exactly equal resource assignment across cortical regions may itself be a costly process; under such a cost, it could be optimal to allow for some variability in resource assignment. Another possibility is that there are unavoidable imperfections in mental inference (*Drugowitsch et al., 2016*) that make it impossible to compute $\bar{J}_{\text{optimal}}$ without error, such that the outcome of the computation will vary from trial to trial even when the stimuli are identical.

## Experimental predictions of incentive manipulations

In the present study, we have focused on effects of set size and probing probability on encoding precision. However, our theory also makes predictions about effects of incentive manipulations on encoding precision, because such manipulations affect the expected behavioral cost function.

Incentives can be experimentally manipulated in a variety of ways. One method used in at least two previously published delayed-estimation experiments is to make the feedback binary ("correct," "error") and vary the value of the maximum error allowed to receive positive feedback (*Zhang and Luck, 2011*; *Nassar et al., 2018*). In both studies, subjects in a "low precision" condition received positive feedback whenever their estimation error was smaller than a threshold value of π/3. Subjects in the "high precision" condition, however, received positive feedback only when the error was smaller than π/12 (*Zhang and Luck, 2011*) or π/8 (*Nassar et al., 2018*). Neither of the two studies found evidence for a difference in encoding precision between the low- and high-precision conditions. At first, this may seem to be at odds with the predictions of our model, as one may expect that it should assign more resource to items in the high-precision condition. To test whether this is the case, we simulated this experimental manipulation using a behavioral cost function $c_{\mathrm{behavioral},i}(\varepsilon)$ that maps values of $|\varepsilon|$ smaller than the feedback threshold to 0 and larger values to 1. The results reveal that the model predictions are not straightforward and that it can actually account for the absence of an effect (*Figure 6*). In particular, the simulation results suggest that the experimental manipulations in the studies by Zhang and Luck and Nassar et al. may not have been strong enough to measure an effect. Indeed, another study has criticized the study by Zhang and Luck on exactly this point and did find an effect when using an experimental design with stronger incentives (*Fougnie et al., 2016*).

Another method to manipulate incentives is to vary the amount of potential reward across items within a display. For example, Klyszejko and colleagues performed a local change discrimination experiment in which the monetary reward for a correct response depended on which item was probed (*Klyszejko et al., 2014*). They found a positive relation between the amount of reward associated with an item and response accuracy, which indicates that subjects spent more resource on encoding items with larger potential reward. This incentive manipulation can be implemented by multiplying the behavioral cost function with an item-dependent factor $u_i$, which modifies *Equation (11)* to

$$\bar{J}_{\mathrm{optimal},i}(r_i;\lambda,\tau) = \underset{\bar{J}}{\arg\min}\left(u_i p_i \bar{c}_{\mathrm{behavioral}}(\bar{J};\tau) + \lambda\bar{J}\right).$$

The coefficients $u_i$ and $p_i$ can be combined into a single "item relevance" coefficient $r_i = u_i p_i$, and all theoretical results and predictions that we derived for $p_i$ now apply to $r_i$.

A difference between the two discussed methods is that the former varied incentives within a trial and the latter across trials. However, both methods can be applied in both ways. A within-trial variant of the experiments by *Zhang and Luck (2011)* and *Nassar et al. (2018)* would be a N = 2 task in which one of the items always has a low positive feedback threshold and the other a high one. Similarly, a

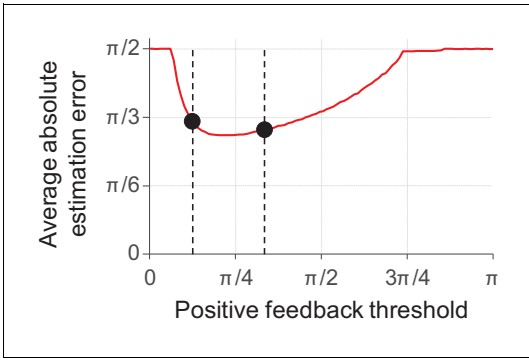

**Figure 6.** Model predictions for a delayed-estimation task with binary feedback (*N* = 5). In this experiment, the observer receives positive feedback (e.g. "correct") when their estimation error is smaller than the positive feedback threshold and negative feedback (e.g. "error") otherwise. We modelled this using a behavioral cost function that maps errors below the feedback threshold to a cost of 0 and errors larger than this threshold to a cost equal to 1. The model predicts that subjects do not invest any resource when the feedback threshold is very small (extremely difficult tasks) or very large (extremely easy tasks), such that the expected absolute estimation error is π/2 (guessing). In an intermediate regime, the prediction is U-shaped and contains a region in which the predicted estimation error barely changes as a function of feedback threshold. In this region, any performance benefit from increasing the amount of invested resource is almost exactly outdone by the added neural cost. The dashed lines show the feedback thresholds corresponding to the "high precision" and "low precision" conditions in the experiment by *Nassar et al. (2018)*. Under the chosen parameter settings (λ = 0.08, τ = 30), the model predicts that the average absolute estimation errors in these two conditions (black circles) are very similar to each other.
DOI: https://doi.org/10.7554/eLife.34963.012

between-trial variant of the experiment by *Klyszejko et al. (2014)* would be to scale the behavioral cost function of items with a factor that varies across trials or blocks, but is constant within a trial. Our model can be used to derive predictions for these task variants, which to our knowledge have not been previously reported in the published literature.

## Neural mechanisms and timescale of optimization

Our results raise the question of what neural mechanism could implement the optimal allocation policy that forms the core of our theory. Some form of divisive normalization (*Bays, 2014*; *Carandini and Heeger, 2012*) would be a likely candidate, which is already a key operation in neural models of attention (*Reynolds and Heeger, 2009*) and visual working memory (*Bays, 2014*; *Wei et al., 2012*). The essence of this mechanism is that it lowers the gain when set size is larger, without requiring explicit knowledge of the set size prior to the presentation of the stimuli. Consistent with the predictions of this theory, empirical work has found that the neural activity associated with the encoding of an item decreases with set size, as observed in for example the lateral intraparietal cortex (*Churchland et al., 2008*; *Balan et al., 2008*) and superior colliculus (*Basso and Wurtz, 1998*). Moreover, the work by *Bays (2014)* has shown that a modified version of divisive normalization can account for the near-optimal distribution of resources across items with unequal probing probabilities. As set size effects in our model are mediated by probing probability, its predicted set size effects can probably be accounted for by a similar mechanism.

Another question concerns the timescale at which the optimization takes place. In all experimental data that we considered here, the only factors that changed from trial to trial were set size (E1-E7) and probing probability (E7-E9). When we fitted the model, we assumed that the expected total cost in these experiments was minimized on a trial-by-trial basis: whenever set size or probing probability changed from one trial to the next, the computation of $\mathbf{J}_{\text{optimal}}$ followed this change. This assumption accounted well for the data and, as discussed above, previous work has shown that divisive normalization can accommodate trial-by-trial changes in set size and probing probability. However, can the same mechanism also accommodate changes in the optimal resource policy changes driven by other factors, such as the behavioral cost function, $c_{\text{behavioral}}(\varepsilon)$? From a computational standpoint, divisive normalization is a mapping from an input vector of neural activities to an output vector, and the shape of this mapping depends on the parameters of the mechanism (such as gain, weighting factors, and a power on the input). As the mapping is quite flexible, we expect that it can accommodate a near-optimal allocation policy for most experimental conditions. However, top-down control and some form of learning (e.g. reinforcement learning) are likely required to adjust the parameters of the normalization mechanism, which would prohibit instantaneous optimality after a change in the experimental conditions.

## Neural prediction

The total amount of resource that subjects spend on item encoding may vary non-monotonically with set size in our model. At the neural level, this translates to a prediction of a non-monotonic relation between population-level spiking activity and set size. We are not aware of any studies that have specifically addressed this prediction, but it can be tested using neuroimaging experiments similar to previously conducted experiments. For example, Balan et al. used single-neuron recording to estimate neural activity per item for set sizes 2, 4, and 6 in a visual search task (*Balan et al., 2008*). To test for the existence of the predicted non-monotonicity, the same recoding techniques can be used in a VWM task with a more fine-grained range of set sizes. Even though it is practically impossible to directly measure population-level activity, reasonable estimates may be obtained by multiplying single-neuron recordings with set size (under the assumption that an increase in resource translates to an increase in firing rate and not an increase of neurons used to encode an item). A similar method can also assess the relation between an item's probing probability and the spiking activity related to its neural encoding.

## Extensions to other domains

Our theory might apply beyond working memory tasks. In particular, it has been speculated that the selectivity of attention arises from a need to balance performance against the costs associated with spiking (*Pestilli and Carrasco, 2005*; *Lennie, 2003*). Our theory provides a normative formalism to

test this speculation and may thus explain set size effects in attention tasks (*Lindsay et al., 1968*; *Shaw, 1980*; *Ma and Huang, 2009*).

Furthermore, developmental studies have found that that working memory capacity estimates change with age (*Simmering and Perone, 2012*; *Simmering, 2012*). Viewed from the perspective of our proposed theory, this raises the question of why the optimal trade-off between behavioral and neural cost would change with age. A speculative answer is that a subject's coding efficiency – formalized by the reciprocal of parameter $\alpha$ in *Equation 7* – may improve during childhood: an increase in coding efficiency reduces the neural cost per unit of precision, which shifts the optimal amount of resource to use for encoding to larger values. Neuroimaging studies might provide insight into whether and how coding efficiency changes with age, for example by estimating the amount of neural activity required per unit of precision in memory representations.

## Broader context

Our work fits into a broader tradition of normative theories in psychology and neuroscience (*Table 4*). The main motivation for such theories is to reach a deeper level of understanding by analyzing a system in the context of the ecological needs and constraints under which it evolved. Besides work on ideal-observer decision rules (*Green and Swets, 1966*; *KordingKörding, 2007*; *Geisler, 2011*; *Shen and Ma, 2016*) and on resource-limited approximations to optimal inference (*Gershman et al., 2015*; *Griffiths et al., 2015*; *Vul and Pashler, 2008*; *Vul, 2009*), normative approaches have also been used at the level of neural coding. For example, properties of receptive fields (*Vincent et al., 2005*; *Liu et al., 2009*; *Olshausen and Field, 1996*), tuning curves (*Attneave, 1954*; *Barlow, 1961*; *Ganguli and Simoncelli, 2010*), neural architecture (*Cherniak, 1994*; *Chklovskii et al., 2002*), receptor performance (*Laughlin, 2001*), and neural network modularity (*Clune et al., 2013*) have been explained as outcomes of optimization under either a cost or a hard constraint (on total neural firing, sparsity, or wiring length), and are thus mathematically closely related to the theory presented here. However, a difference concerns the timescale at which the optimization takes place: while optimization in the context of neural coding is typically thought to take place at the timescale over which the statistics of the environment change or a developmental timescale, the theory that we presented here could optimize on a trial-by-trial basis to follow changes in task properties.

We already mentioned the information-theory models of working memory developed by Chris R. Sims et al. A very similar framework has been proposed by Chris A. Sims in behavioral economics, who used information theory to formalize his hypothesis of "rational inattention," that is, the hypothesis that consumers make optimal decisions under a fixed budget of attentional resources that can be allocated to process economic data (*Sims, 2003*). The model presented here differs from these two approaches in two important ways. First, similar to early models of visual working memory limitations, they postulate a fixed total amount of resources (formalized as channel capacity), which is a constraint rather than a cost. Second, even if it had been a cost, it would have been the expected value of a log probability ratio. Unlike neural spike count, a log probability ratio does not obviously map to a biologically meaningful cost on a single-trial level. Nevertheless, recent work has

**Table 4.** Examples of resource-rationality concepts in neuroscience, psychology, and economics.

| Study | Optimized quantity | Performance term | Resource cost/constraint |
|---|---|---|---|
| Efficient coding in neural populations | | | |
| *Ganguli and Simoncelli (2010)* | Tuning curve spacing and width | Fisher information or discriminability | Neural activity (constraint) |
| *Olshausen and Field (1996)* | Receptive field specificity | Information | Sparsity |
| Capacity "limitations" in attention and memory | | | |
| *Sims et al. (2012)* | Information channel bit allocation | Channel distortion (e.g. squared error) | Channel capacity (constraint) |
| Van den Berg and Ma (present study) | Mean encoding precision | Behavioral task accuracy | Neural activity (cost) |
| Rational inattention in consumer choice | | | |
| *Sims (2003)* | Distribution of attention | Channel distortion (e.g. squared error) | Channel capacity (constraint) |

DOI: https://doi.org/10.7554/eLife.34963.013

attempted to bridge rational inattention and attention in a psychophysical setting (*Caplin et al., 2018*).

## Materials and methods

### Data and code sharing

Data from experiments E1-E7 (Table 1) and Matlab code for model fitting and simulations are available at http://dx.doi.org/10.5061/dryad.nf5dr6c.

### Statistical analyses

Bayesian t-tests were performed using the JASP software package (*JASP Team, 2017*) with the scale parameter of the Cauchy prior set to its default value of 0.707.

### Model fitting

We used a Bayesian optimization method (*Acerbi and Ma, 2017*) to find the parameter vector $\theta = \{\beta, \lambda, \tau\}$ that maximizes the log likelihood function, $\sum_{i=1}^{n} logp(\varepsilon_i; p_i, \theta)$, where $n$ is the number of trials in the subject's data set, $\varepsilon_i$ the estimation error on the $i$th trial, and $p_i$ the probing probability of the probed item on that trial. To reduce the risk of converging into a local maximum, initial parameter estimates were chosen based on a coarse grid search over a large range of parameter values. The predicted estimation error distribution for a given parameter vector $\theta$ and probing probability $p_i$ was computed as follows. First, $\bar{J}_{\text{optimal}}$ was computed by applying Matlab's fminsearch function to *Equation 11*. Thereafter, the gamma distribution over $J$ (with mean $\bar{J}_{\text{optimal}}$ and shape parameter $\tau$) was discretized into 50 equal-probability bins. The predicted (Von Mises) estimation error distribution was then computed under the central value of each bin. Finally, these 50 predicted distributions were averaged. We verified that increasing the number of bins used in the numerical approximation of the integral over $J$ did not substantially affect the results.

### Model comparison using cross-validation

In the cross-validation analysis, we fitted the models in the same way as described above, but using only 80% of the data. We did this five times, each time leaving out a different subset of 20% of the data (in the first run we left out trials 1, 6, 11; in the second run we left out trials 2, 7, 12, etc.). At the end of each run, we used the maximum-likelihood parameter estimates to compute the log likelihood of the 20% of trials that were left out. These log likelihood values were then combined across the five runs to give an overall cross-validated log likelihood value for each model.

## Acknowledgments

This work was funded by grant R01EY020958 from the National Institutes of Health, grant 2015–00371 by the Swedish Resarch Council, and grant INCA 600398 by Marie Sklodowska Curie Actions. We thank all authors of the papers listed in Table 1 for making their data available.

## Additional information

### Funding

| Funder | Grant reference number | Author |
|---|---|---|
| Vetenskapsrådet | 2015-00371 | Ronald van den Berg |
| Marie Skłodowska-Curie Actions, Cofund | INCA 600398 | Ronald van den Berg |
| National Institutes of Health | R01EY020958 | Wei Ji Ma |

The funders had no role in study design, data collection and interpretation, or the decision to submit the work for publication.

## Author contributions
Ronald van den Berg, Conceptualization, Formal analysis, Investigation, Methodology, Writing—original draft, Writing—review and editing; Wei Ji Ma, Conceptualization, Data curation, Formal analysis, Supervision, Visualization, Methodology, Writing—original draft

## Author ORCIDs
Ronald van den Berg http://orcid.org/0000-0001-7353-5960
Wei Ji Ma https://orcid.org/0000-0002-9835-9083

## Decision letter and Author response
Decision letter https://doi.org/10.7554/eLife.34963.022
Author response https://doi.org/10.7554/eLife.34963.023

## Additional files

### Supplementary files
• Transparent reporting form
DOI: https://doi.org/10.7554/eLife.34963.014

### Data availability
Data from experiments E1-E7 (Table 1) and Matlab code for model fitting and simulations are available at http://dx.doi.org/10.5061/dryad.nf5dr6c.

The following dataset was generated:

| Author(s) | Year | Dataset title | Dataset URL | Database, license, and accessibility information |
| --- | --- | --- | --- | --- |
| Ronald van den Berg, Wei Ji Ma | 2018 | Data from: A resource-rational theory of set size effects in human visual working memory | http://dx.doi.org/10.5061/dryad.nf5dr6c | Available at Dryad Digital Repository under a CC0 Public Domain Dedication |

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

## Appendix 1

DOI: https://doi.org/10.7554/eLife.34963.015

## Relation between Fisher information *J* and concentration parameter $\kappa$

As we are only considering stimuli with circular domains, we assume that memory encoding errors follow a Von Mises distribution with a concentration parameter $\kappa$,

$$p(\varepsilon|\kappa) = \frac{1}{2\pi I_0(\kappa)}e^{\kappa cos(\varepsilon)}, \tag{15}$$

where $I_0$ is the modified Bessel function of the first kind of order 0. We measure encoding precision as Fisher information, *J*, which measures the performance of the best possible unbiased decoder. Substituting *Equation 15* into the definition of Fisher information, we find that *J* and $\kappa$ are one-to-one related through

$$J = \kappa\frac{I_1(\kappa)}{I_0(\kappa)} \tag{16}$$

Encoding precision *J* is a monotonically increasing function of $\kappa$ and therefore invertible. However, the inverse is not analytic, so we use numerical inversion to compute the mapping from *J* to $\kappa$ when fitting models.

## Mathematical proofs of some properties of the resource-rational model for local tasks

In this section, we prove three properties of the general model that we presented for "local" tasks, that is, tasks in which responses depend on a single item. This model is characterized by *Equation 11*,

$$\bar{J}_{\text{optimal},i}(p_i;\lambda) = \underset{\bar{J}}{\text{argmin}}\,(p_i\bar{c}_{\text{behavioral}}(\bar{J}) + \lambda\bar{J}), \tag{17}$$

where $\bar{J} \geq 0$, $p_i \in [0,1]$, and $\lambda \geq 0$, and we left out the dependence on the parameter $\tau$ for notational convenience. We will also use the derivative of the local expected total cost,

$$\bar{c}_{\text{total}}(\bar{J}) = p_i\bar{c}_{\text{behavioral}}{}'(\bar{J}) + \lambda, \tag{18}$$

where $\bar{c}_{\text{behavioral}}$ is the derivative of the expected behavioral cost.

We will now prove that the following three claims hold under rather general assumptions about the shape of the expected behavioral cost function in this model:

*Claim 1.* When neural coding is costly ($\lambda > 0$), it is optimal to encode items with a finite amount of resource;

*Claim 2.* It is sometimes optimal not to encode a task-relevant item;

*Claim 3.* When each item is equally likely to be probed, $p_i = 1/N$, the optimal amount of resource per item decreases with set size.

### Assumptions about the expected behavioral cost

We construct our proofs under two intuitive and general assumptions about the expected behavioral cost function $\bar{c}_{\text{behavioral}}(\bar{J})$:

*Assumption 1.* Expected behavioral cost is a monotonically decreasing function of resource: whenever more resource is invested, the expected behavioral cost is lower. This means that $\bar{c}_{\text{behavioral}}(\bar{J}) \leq 0$ for all $\bar{J}$.

*Assumption 2.* A law of diminishing returns: when adding a bit of extra resource, the resulting decrease in $\bar{c}_{\text{behavioral}}(\bar{J})$ is lower in magnitude when $\bar{J}$ is higher. This means that

$\bar{c}_{\text{behavioral}}{}'(\bar{J})$ is monotonically increasing, that is, $\bar{c}_{\text{behavioral}}{}''(\bar{J})$ for all $\bar{J}$. As a consequence, $\bar{c}_{\text{behavioral}}{}'(\bar{J})$ takes its lowest value at $\bar{J}=0$ and its largest as $\bar{J}\to\infty$.

Both assumptions are satisfied by the behavioral cost function that we used for fitting human data, namely $c_{\text{behavioral}}(\varepsilon;\beta)=|\varepsilon|^{\beta}$. Examples of the expected behavioral cost function under this choice and its first and second derivative are presented in *Appendix 1—figure 1*.

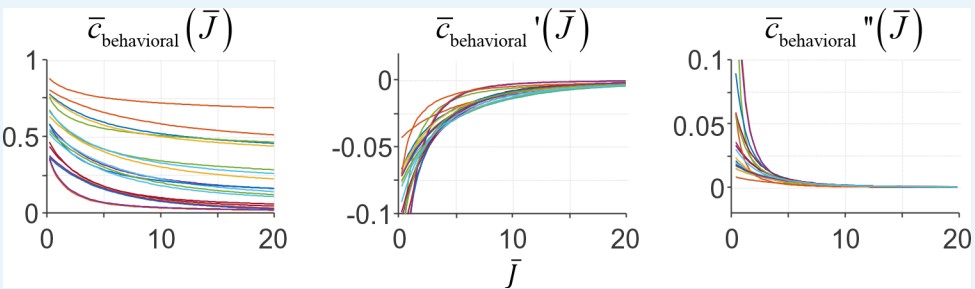

**Appendix 1—figure 1.** Examples of the expected behavioral cost function and its first and second derivative under a behavioral cost function $c_{\text{behavioral}}(\varepsilon)=|\varepsilon|^{\beta}$. Different colors represent different choices of parameters $\beta$ and $\tau$ (randomly drawn).
DOI: https://doi.org/10.7554/eLife.34963.016

## Three scenarios

We now return to the problem of calculating $\bar{J}_{\text{optimal}}$, *Equation 17*. We are interested in the value $\bar{J}\in[0,\infty)$ that minimizes the expected total cost, $\bar{c}_{\text{total}}(\bar{J})$. We separately consider the following three scenarios: the minimum lies on the left boundary (0), on the right boundary ($\infty$), or in between.

## *Scenario 1: $\bar{c}_{\text{total}}(\bar{J})$ is monotonically decreasing across the domain of $\bar{J}$, so $\bar{J}_{\text{optimal}}\to\infty$.*

When does this happen? The monotonic decrease means that $\bar{c}_{\text{total}}{}'(\bar{J})\leq 0$ for all $\bar{J}$, or equivalently, $p_i\bar{c}_{\text{behavioral}}{}'(\bar{J})\leq-\lambda$ for all $\bar{J}$. As we assume $\bar{c}_{\text{behavioral}}{}'(\bar{J})$ to be monotonically increasing (Assumption 2), its largest value is attained at $\bar{J}\to\infty$. Therefore, $p_i\bar{c}_{\text{behavioral}}{}'(\bar{J})\leq-\lambda$ is equivalent to $p_i\bar{c}_{\text{behavioral}}(\infty)\leq-\lambda$, or (using Assumption 1) $p_i|\bar{c}_{\text{behavioral}}{}'(\infty)|\geq\lambda$. This means that it is optimal to invest infinite resource when $p_i$ exceeds a critical value $p_\infty$:

$$\bar{J}_{\text{optimal}}=\infty \text{ when } p_i\geq p_\infty\equiv\frac{\lambda}{|\bar{c}_{\text{behavioral}}{}'(\infty)|}.$$

The condition $p_i\geq p_\infty$ is satisfied when $\lambda=0$. This makes sense: when neural cost plays no role, there is no reason not to invest more. Other than that, the condition will rarely if ever be satisfied, as every expected behavioral cost function that we can think of has the property $|\bar{c}_{\text{behavioral}}{}'(\infty)|=0$: as the amount of invested resource approaches infinity, there is no behavioral benefit in investing more resource (note that $p_\infty$ has a domain $[0,\infty)$, not $[0,1]$). Therefore, unless neural cost plays no role, we do not expect it to be optimal to invest an infinite amount of resource in an item.

In tasks where $p_i$ is one-to-one related to set size, the above result can be reformulated in terms of set size. In particular, when probing probabilities are equal, $p_i=\frac{1}{N}$, the above result implies that there exists a set size $N_\infty$ (in general not an integer) below which it is optimal to invest infinite resource in each item:

$$\bar{J}_{\text{optimal}}=\infty \text{ when } N\leq N_\infty\equiv\frac{1}{p_\infty}=\frac{|\bar{c}_{\text{behavioral}}{}'(\infty)|}{\lambda}$$

### Scenario 2: $\bar{c}_{\text{total}}(\bar{J})$ is monotonically increasing across the domain of $\bar{J}$, so $\bar{J}_{\text{optimal}} = 0$.

The monotonic increase means that $\bar{c}_{\text{total}}'(\bar{J}) \geq 0$ for all $\bar{J}$, or equivalently, $p_i\bar{c}_{\text{behavioral}}'(\bar{J}) \geq -\lambda$ for all $\bar{J}$. As we assume $\bar{c}_{\text{behavioral}}'(\bar{J})$ to be monotonically increasing (Assumption 2), its smallest value is attained at $\bar{J} \to \infty$. Therefore, $p_i\bar{c}_{\text{behavioral}}'(\bar{J}) \geq -\lambda$ is equivalent to $p_i\bar{c}_{\text{behavioral}}'(0) \geq -\lambda$, or (using Assumption 1) $p_i|\bar{c}_{\text{behavioral}}'(0)| \leq \lambda$. This means that it is optimal to invest no resource when $p_i$ is smaller than or equal to a critical value $p_0$:

$$\bar{J}_{\text{optimal}} = 0 \text{ when } p_i \leq p_0 \equiv \frac{\lambda}{|\bar{c}_{\text{behavioral}}'(0)|}$$

A similar condition was derived in our earlier work (*de Silva and Ma, 2018*) for the case of a fixed total amount of resource (hard constraint).

The condition $p_i \leq p_0$ is satisfied when $p_i = 0$. This makes sense: when an item never gets probed, one should not invest any resource. More generally, when probing probability is sufficiently low, the behavioral cost function is sufficiently shallow at 0, and neural cost is sufficiently important, it is not worth investing any resource on encoding. The expression for $p_0$ also makes clear that the optimal amount of resource is never 0 when the slope of the behavioral cost function at 0 approaches $-\infty$.

In tasks where $p_i$ is one-to-one related to set size, the above result can be reformulated in terms of set size. In particular, when probing probabilities are equal, $p_i = \frac{1}{N}$, the above result implies that there exists a set size $N_0$ (in general not an integer) beyond which it is optimal to not invest any resource in any item:

$$\bar{J}_{\text{optimal}} = 0 \text{ when } N \geq N_0 \equiv \frac{1}{p_0} = \frac{|\bar{c}_{\text{behavioral}}'(0)|}{\lambda}.$$

Intuitively, this means that when set size is too large, the chances of success are too low and one should not even try.

### Scenario 3: $\bar{c}_{\text{total}}(\bar{J})$ has a stationary point, so $\bar{J}_{\text{optimal}}$ is finite and nonzero.

We will now consider the remaining scenario, which is the complement of Scenarios 1 and 2; in particular, we can take $\lambda > 0$ and $p_i > 0$. The stationary point of $\bar{c}_{\text{total}}(\bar{J})$ will always be a minimum, as the second derivative $\bar{c}_{\text{total}}''(\bar{J})$ is equal to $\bar{c}_{\text{behavioral}}''(\bar{J})$, which is always positive (Assumption 2). At the minimum, we have $\bar{c}_{\text{total}}(\bar{J}) = 0$, from which it follows that $\bar{c}_{\text{behavioral}}'(\bar{J}) = -\frac{a}{p_i}$ at the minimum. As the left-hand side is monotonically increasing as a function of $\bar{J}$ (Assumption 2), the minimum is either a single point or a single interval, but there cannot be multiple disjoint minima. Graphically, this equation describes the intersection between $\bar{c}_{\text{behavioral}}'(\bar{J})$, which is a monotonically increasing function, and a flat line at a value $-\frac{\lambda}{p_i}$ (*Appendix 1—figure 2*). The value of at which this intersection occurs necessarily increases with $p_i$.

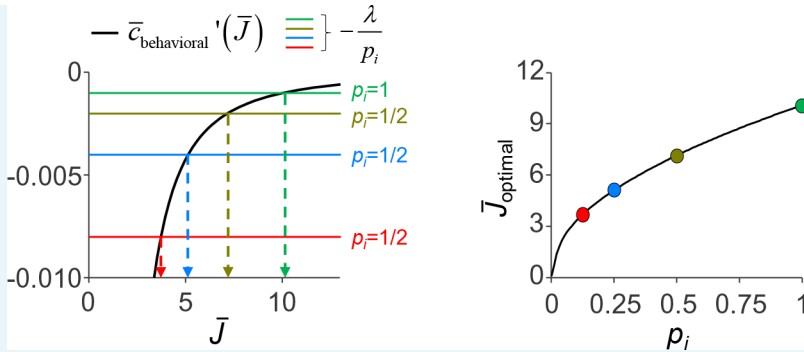

**Appendix 1–figure 2.** Graphical illustration of the solution to the cost-minimization problem that determines the value of $\bar{J}_{\text{optimal}}$. The cost-minimizing value of solution of $\bar{J}$ lies at the intersection between the derivative of the expected behavioral cost function (black curve) and a flat line at a value $-\lambda/p_i$ (colored lines). This value (indicated with arrows) necessarily increases with $p_i$. The parameter values used in this simulation were the same as those used to generate **Figure 2D and E** ($\lambda = 0.01$, $\beta = 2$, $\tau{\downarrow}0$).
DOI: https://doi.org/10.7554/eLife.34963.017

## Three regimes for probing probability

So far, we have assumed a given probing probability $p_i$. Now suppose that for a given $\bar{c}_{\text{behavioral}}(\bar{J})$ and a given $\lambda$, we increase $p_i$ from 0 to 1:

- The first regime is $p_i \leq p_0$. There, Scenario 2 applies and $\bar{J}_{\text{optimal}} = 0$: the item does not get encoded at all.
- The second regime is $p_0 < p_i < p_\infty$; there, Scenario 3 applies and $\bar{J}_{\text{optimal}}$ monotonically increases with $p_i$.
- The third regime is $p_i \geq p_\infty$. There, Scenario 1 applies and $\bar{J}_{\text{optimal}} = \infty$: the item gets encoded with infinite resource.

Even though not all regimes might exist for every parameter combination, the model generally predicts that there is a regime in which $\bar{J}_{\text{optimal}}$ increases monotonically with $p_i$ (**Figure 1D**).

## Three regimes for set size

We can similarly examine the experimentally important special case of equal probing probabilities, $p_i = \frac{1}{N}$:

The first regime is $N \leq N_\infty$. There, Scenario 1 applies and $\bar{J}_{\text{optimal}} = \infty$: all items are encoded with infinite resource.

The second regime is $N_\infty < N < N_0$. There, Scenario 3 applies and $\bar{J}_{\text{optimal}}$ monotonically decreases with $N$.

The third regime is $N \geq N_0$. There, Scenario 2 applies and $\bar{J}_{\text{optimal}} = 0$: no items are encoded at all.

Even though not all regimes might exist for every parameter combination, the model generally predicts that there is a regime in which $\bar{J}_{\text{optimal}}$ decreases monotonically with $N$ (**Figure 1E**).

## Conclusion

In conclusion, given **Equation (17)** and two additional assumptions, we have proven the following:

- Investing infinite resource in an item is only optimal when $p_i|\bar{c}_{\text{behavioral}}{}'(\infty)| \geq \lambda$. In practice, this might only happen when neural cost is unimportant ($\lambda = 0$). This proves Claim 1.

- Investing no resource in an item is optimal when $p_i|\bar{c}_{\text{behavioral}}{}'(0)| \leq \lambda$. This can happen even when the probing probability $p_i$ is nonzero. This proves Claim 2.
- $\bar{J}_{\text{optimal}}$ is a monotonically increasing function of $p_i$. In particular, if $p_i = \frac{1}{N}$, then $\bar{J}_{\text{optimal}}$ is a monotonically decreasing function of $N$. This proves Claim 3.

All three results hold more generally than we have shown here: we can replace the neural cost term $\lambda\bar{J}$ in **Equation (17)** by any function $c_{\text{neural}}(\bar{J})$ whose derivative is positive and monotonically increasing. The proofs proceed along the same lines (see below).

## Special case: fixed-precision model

For the fixed-precision model (variable-precision model with $\tau \downarrow 0$), **Equation (12)** in the main text takes the form

$$\bar{c}_{\text{behavioral}}(J) = \int\limits_{-\pi}^{\pi} c_{\text{behavioral}}(\varepsilon) VM(\varepsilon; J)d\varepsilon.$$

We wish to evaluate $p_0 \equiv \frac{\lambda}{|\bar{c}_{\text{behavioral}}(0)|}$. First, we evaluate the derivative of $\bar{c}_{\text{behavioral}}(J)$ using the chain rule:

$$\frac{d\bar{c}_{\text{behavioral}}}{dJ} = \frac{d\bar{c}_{\text{behavioral}}}{d\kappa}\frac{d\kappa}{dJ}. \tag{19}$$

Using **Equation (15)**, the first factor is

$$\begin{aligned}
\frac{d\bar{c}_{\text{behavioral}}}{d\kappa} &= \frac{d}{d\kappa}\int\limits_{-\pi}^{\pi} c_{\text{behavioral}}(\varepsilon)\frac{1}{2\pi I_0(\kappa)}e^{\kappa\cos(\varepsilon)}d\varepsilon \\
&= \frac{1}{2\pi}\int\limits_{-\pi}^{\pi} c_{\text{behavioral}}(\varepsilon)\frac{d}{d\kappa}\left(\frac{1}{I_0(\kappa)}e^{\kappa\cos(\varepsilon)}\right)d\varepsilon \\
&= \frac{1}{2\pi}\int\limits_{-\pi}^{\pi} c_{\text{behavioral}}(\varepsilon)\frac{d}{d\kappa}\left(-\frac{I_0{}'(\kappa)}{I_0(\kappa)^2}e^{\kappa\cos(\varepsilon)} + \frac{\cos(\varepsilon)}{I_0(\kappa)}e^{\kappa}\right)d\varepsilon \\
&= \frac{1}{2\pi}\int\limits_{-\pi}^{\pi} c_{\text{behavioral}}(\varepsilon)\frac{d}{d\kappa}\left(-\frac{I_1(\kappa)}{I_0(\kappa)^2}e^{\kappa\cos(\varepsilon)} + \frac{\cos(\varepsilon)}{I_0(\kappa)}e^{\kappa}\right)d\varepsilon
\end{aligned} \tag{20}$$

where in the last line we used $I_0(\kappa) = I_1(\kappa)$ (see Eq. 9.6.27 in **Abramowitz and Stegun [1972]**). We next evaluate the second factor in **Equation (19)** using **Equation (16)**:

$$\begin{aligned}
\frac{d\kappa}{dJ} &= \left(\frac{dJ}{d\kappa}\right)^{-1} = \left(\frac{d}{d\kappa}\left(\frac{\kappa I_1(\kappa)}{I_0(\kappa)}\right)\right)^{-1} = \left(\frac{ppaI_0(\kappa)^2 - \kappa I_1(\kappa)I_0(\kappa)}{I_0(\kappa)^2}\right)^{-1} \\
&= \left(\kappa\left(1 - \frac{I_1(\kappa)^2}{I_0(a)^2}\right)\right)^{-1},
\end{aligned} \tag{21}$$

where in the third equality, we used $\frac{d}{d\kappa}(\kappa I_1(\kappa)) = \kappa I_0(\kappa)$ (see Eq. 9.6.28 in [Abramowitz & Stegun, 1972]). We now combine **Equation (20)** and **Equation (21)** into **Equation (19)** and the result in turn in the expression for $p_0$. We also realize that the limit $J \downarrow 0$ is, using **Equation (16)**, equivalent to the limit 0. Putting everything together, we find

$$p_0 = \lambda \lim_{\kappa\downarrow 0}\left|\frac{\kappa\left(1 - \frac{I_1(\kappa)^2}{I_0(\kappa)^2}\right)}{\frac{1}{2\pi}\int\limits_{-\pi}^{\pi} c_{\text{behavioral}}(\varepsilon)\left(-\frac{I_1(\kappa)}{I_0(\kappa)^2}e^{\kappa\cos\varepsilon} + \frac{\cos\varepsilon}{I_0(\kappa)}e^{\kappa}\right)d\varepsilon}\right| = 0.$$

We conclude that in our theory for delayed-estimation, assuming the expected behavioral cost function from the fixed-precision model, it is only optimal to invest no resource at all into an item when that item has zero probability of being probed.

## Generalization to other neural cost functions

So far, we have assumed that the expected neural cost is linear in resource, **Equation (8)**. Relaxing this assumption, **Equation (17)** for local tasks becomes

$$\bar{J}_{\text{optimal},i}(p_i;\lambda) = \underset{\bar{J}}{\arg\min}\left(p_i\bar{c}_{\text{behavioral}}(\bar{J}) + \lambda\bar{c}_{\text{neural}}(\bar{J})\right).$$

The derivative of the local expected total cost becomes

$$\bar{c}_{\text{total}}(\bar{J}) = p_i\bar{c}_{\text{behavioral}}{}'(\bar{J}) + \lambda\bar{c}_{\text{neural}}{}'(\bar{J}).$$

The three claims above still hold if we modify the two assumptions to

*Assumption 1'.* $\frac{\bar{c}_{\text{behavioral}}{}'(\bar{J})}{\bar{c}_{\text{neural}}{}'(\bar{J})} \leq 0$ for all $\bar{J}$.

*Assumption 2'.* $\frac{\bar{c}_{\text{behavioral}}{}'(\bar{J})}{\bar{c}_{\text{neural}}{}'(\bar{J})}$ is monotonically increasing for all $\bar{J}$.

The proofs are completely analogous, with $\bar{c}_{\text{behavioral}}(\bar{J})$ replaced by $\frac{\bar{c}_{\text{behavioral}}{}'(\bar{J})}{\bar{c}_{\text{neural}}{}'(\bar{J})}$.

## Optimal decision rule for the change detection task

In our simulation of the change detection task, we assume that observers use a Bayesian decision rule. This rule is to report "change" whenever the posterior ratio of change presence over change absence exceeds 1,

$$\frac{p(\text{changepresent}|\mathbf{x},\mathbf{y})}{p(\text{changeabsent}|\mathbf{x},\mathbf{y})} > 1,$$

where **x** and **y** denote the vectors of noisy measurements of the items in the first and second displays, respectively. Under the Von Mises noise assumption, and assuming a flat prior on change presence, this decision rule evaluates to (*Keshvari et al., 2013*)

$$\frac{1}{N}\sum_{i=1}^{N}\frac{I_0\left(\kappa_{\text{x},i}\right)I_0\left(\kappa_{\text{y},i}\right)}{I_0\left(\sqrt{\kappa_{\text{x},i}^2 + \kappa_{\text{x},i}^2 + 2\kappa_{\text{x},i}\kappa_{\text{y},i}cos(y_i - x_i)}\right)} > 1,$$

where $\kappa_{\text{x},i}$ and $\kappa_{\text{y},i}$ denote the concentration parameters of the Von Mises distributions associated with the observations of the items at the $i^{\text{th}}$ location in the first and second displays, respectively. The predicted probability of a correct response for a given resource vector, $p(\text{error}|\bar{\mathbf{J}})$, is not analytic, but can easily be computed using Monte Carlo simulations.

