## [Decision Letter]

Thank you for submitting your article "Ecological rationality in human working memory and attention" for consideration by *eLife*. Your article has been reviewed by three peer reviewers, and the evaluation has been overseen by a Reviewing Editor and Timothy Behrens as the Senior Editor. The following individual involved in review of your submission has agreed to reveal her identity: Jacqueline Gottlieb (Reviewer #2).

The reviewers have discussed the reviews with one another and the Reviewing Editor has drafted this decision to help you prepare a revised submission.

Summary:

In this manuscript, van den Berg and Ma propose a normative theory of a phenomenon called set-size effect – the fact that attention and working memory performance degrades as a function of the number of items. The set-size effect is fundamental in attention and working memory research, and many previous models have successfully described it by using ad hoc assumptions about the relation between precision and set-size. The authors' main innovation is to explain this relation in a normative framework. Specifically, they propose that the degradation in precision as a function of set size reflects a tradeoff between the benefits of precise encoding and the neural costs that this encoding demands. The authors develop the model, use it to fit several existing data sets, and offer an extensive discussion of the limitations of their model and its relation to the previous theoretical and empirical literatures.

The paper incorporates a genuinely new idea and is clearly written and quite thorough in its analysis and discussion. The Results section details a substantive amount of work, in which the model was fit to data from several experiments where working memory precision was measured in the context of local delayed-estimation, global estimation, change detection, change localization and present/absent visual search, and was quantitatively evaluated against other (often equally good) prediction schemes.

Some major revisions are requested to clarify situations in which not all items are stored, to answer questions about the model parameter τ, and to expand the discussion about how the flexibility of resource allocation in the model could be mechanistically realized in the brain.

Essential revisions:

1) The number of encoded items: The paper mentions a possible hard constraint on the number of items encoded. Depending on set size and cost functions, when (if ever) is it "optimal" not to encode some of the items? Some new modeling results should be shown here to shore this up and a longer discussion of this point should be added to the Results and Discussion.

Furthermore, please expand the discussion of the costs of encoding more items versus fewer. Intuitively, it is obvious that it is more costly to encode 8 items than 2, but there are numerous reasons why this could be the case. Are there any experimental data in support of this assumption? In particular, are there any experimental data showing that encoding more items results in higher firing rates (at the population level)?

Michael Frank and his group have recently made a very principled attempt to characterize the nature of the cost in these tasks, in terms of how participants might group or chunk the items. It may be beyond the scope of the current paper to attempt to outline a theory that explains these results, but perhaps readers would enjoy some more elaborate discussion of this issue.

2) τ: Please provide additional model results that show how the fits look when τ = 0. In particular, please show the goodness of fit of the rational model (with τ = 0) as compared to the fit of a model with a "hard" constraint on resources? In general, it was confusing that the theory is initially described in absence of τ, while τ is used for actual fits to data. It would have been easier to understand if the theory had been evaluated in presence of τ, and its effect studied within that theoretical framework.

3) The speed of policy update: An assumption of the model that may be problematic is that people must almost instantaneously optimize their encoding precision when set-sizes change unpredictably from trial to trial. The theory predicts that, when a trial contains 2 stimuli and the next trial contains 4 stimuli, the participant instantaneously lowers the encoding precision to the new (near-) optimal level. This process sounds pretty demanding itself, especially considering that humans may also play an active role in determining their intrinsic motivation or deciding which items to memorize, which may further slow down the adjustment process. The flexibility in allocating resources that is implied by this model seems to be at odds with the slowness of cognitive control, well-documented by task switching costs. The authors touch on this point in the very last line of the paper, where they note that divisive normalization can provide a rapid adjustment mechanism. Even though the discussion is already long, it would be good to hear more about this point, and a comparison between a hardwired allocation mechanism and slower but more flexible cognitive control strategies.

Furthermore, in the Frank paper described above, they use a task in which there is binary feedback that depends on the liberal vs. conservative error criterion (and they don't report major differences in performance as a function of this). One might suspect that participants would fail to adapt their policy even in an incentive-compatible version of this task which systematically varied these behavioral costs, and this would present a challenge to the authors' theory as described here. Please add some text to the Discussion addressing this point.

4) Total precision vs. set-size: A novel and interesting prediction of the model is the non-monotonic relation between set-size and total precision (Figure 3B). Although the authors state that this point requires more empirical documentation, are the model results consistent with a non-monotonic encoding of target location that was reported by Balan et al. (2008) in monkey area LIP? That study found that, in a covert visual search task with different set sizes, the fidelity of target location encoding by area LIP was higher at set size 4 than at set size 2 or set size 6 (see Figure 5) – a non-linearity that was puzzling at the time but may gain new significance in light of this paper. Please add some discussion of this point to the manuscript.

---

## [Author Response]

Essential revisions:1) The number of encoded items: The paper mentions a possible hard constraint on the number of items encoded. Depending on set size and cost functions, when (if ever) is it "optimal" not to encode some of the items? Some new modeling results should be shown here to shore this up and a longer discussion of this point should be added to the Results and Discussion.

This is an interesting question, because its answer can possibly provide a principled bridge between slot-based and resource models of VWM. We now address this question in three different places. First, a mathematical analysis of the general conditions under which it is optimal to not encode an item is provided in Appendix 1. Second, the question is addressed in the context of delayed-estimation tasks in the new Results section “Is it ever optimal to not encode an item?”.Third, for the change-detection task, the question is addressed in the new section “Predictions for a global task: whole-display change detection”.

Furthermore, please expand the discussion of the costs of encoding more items versus fewer. Intuitively, it is obvious that it is more costly to encode 8 items than 2, but there are numerous reasons why this could be the case. Are there any experimental data in support of this assumption? In particular, are there any experimental data showing that encoding more items results in higher firing rates (at the population level)?

For many choices of spike variability, the total precision of a set of stimuli encoded in a neural population is proportional to the trial-averaged neural spiking rate (e.g., Paradiso, 1988; Seung and Sompolinsky, 1993; Ma et al., 2006). Based on this theoretical argument, it is expected that it is more costly (in terms of neural spiking) to encode 8 items compared to 2, *if they are encoded with the same precision.*

However, it is important to keep in mind that our model does not predict that the total spiking rate will increase with set size, because it generally predicts the precision per item (i.e., spike rate per item) to decrease with set size, which is consistent with physiological evidence (e.g., Churchland et al., 2008; Balan et al., 2008; Basso and Wurtz, 1998). The maximum-likelihood fits suggest that the total amount of invested resource varies non-monotonically with set size, which predicts that the population-level spiking activity also varies non-monotonically with set size. We are not aware of any work that strongly supports or rejects this prediction (see also our response below to the point about the Balan et al. paper). We address this point as follows in a new discussion section “Neural prediction”.

Michael Frank and his group have recently made a very principled attempt to characterize the nature of the cost in these tasks, in terms of how participants might group or chunk the items. It may be beyond the scope of the current paper to attempt to outline a theory that explains these results, but perhaps readers would enjoy some more elaborate discussion of this issue.

We assume that this comment refers to the recent paper by Nassar, Helmers, and Frank. If we understand correctly, this paper is currently in press, so we base our response to this comment on the preprint that is available on bioRxiv.

We agree that this paper has several connections with our own study, and we now refer to it at two different places. First, in the Introduction:

“Finally, Nassar and colleagues have proposed a normative model in which a strategic trade-off is made between the number of encoded items and their precision: when two items are very similar, they are encoded as a single item, such that there is more resource available per encoded item (Nassar et al., 2018). […] However, just as in much of the work discussed above, this theory assumes a fixed resource budget for item encoding, which is not necessarily optimal when resource usage is costly.”

And then again in the “Limitations” section in the Discussion:

“A final limitation is that our theory assumes that items are uniformly distributed and uncorrelated. […] Hence, it seems worthwhile to examine models that combine resource rationality with chunking.”

2) τ: Please provide additional model results that show how the fits look when τ = 0. In particular, please show the goodness of fit of the rational model (with τ = 0) as compared to the fit of a model with a "hard" constraint on resources?

We have added this analysis:

“So far, we have assumed that there is random variability in the actual amount of resource assigned to an item. […] Therefore, we will only consider variable-precision models in the remainder of the paper.”

As in the variable-precision model, the optimal amount of resource per item decreases with set size in the equal-precision variant of the rational model:

Since we think that focusing too much on equal-precision results distracts a bit from the main story, we decided not to include this plot in the paper. As we explain in response to a later comment, the difference between the equal-precision and variable-precision models is mainly in the predicted kurtosis (“peakiness”) of the error distribution, not in the variance of these distributions (let alone in how the variance changes with set size). Hence, the equal-precision vs.variable-precision question is orthogonal to our main question.

In general, it was confusing that the theory is initially described in absence of τ, while τ is used for actual fits to data. It would have been easier to understand if the theory had been evaluated in presence of τ, and its effect studied within that theoretical framework.

Sorry, this was indeed confusing. In the rewritten “Theory” section, we explicitly indicate which equations depend on τ, by including it in the function arguments.

3) The speed of policy update: An assumption of the model that may be problematic is that people must almost instantaneously optimize their encoding precision when set-sizes change unpredictably from trial to trial. The theory predicts that, when a trial contains 2 stimuli and the next trial contains 4 stimuli, the participant instantaneously lowers the encoding precision to the new (near-) optimal level. This process sounds pretty demanding itself, especially considering that humans may also play an active role in determining their intrinsic motivation or deciding which items to memorize, which may further slow down the adjustment process. The flexibility in allocating resources that is implied by this model seems to be at odds with the slowness of cognitive control, well-documented by task switching costs. The authors touch on this point in the very last line of the paper, where they note that divisive normalization can provide a rapid adjustment mechanism. Even though the discussion is already long, it would be good to hear more about this point, and a comparison between a hardwired allocation mechanism and slower but more flexible cognitive control strategies.

This is an important issue, which we now discuss in the new Discussion section “Neural mechanisms and timescale of optimization”.

Furthermore, in the Frank paper described above, they use a task in which there is binary feedback that depends on the liberal vs. conservative error criterion (and they don't report major differences in performance as a function of this). One might suspect that participants would fail to adapt their policy even in an incentive-compatible version of this task which systematically varied these behavioral costs, and this would present a challenge to the authors' theory as described here. Please add some text to the Discussion addressing this point.

The experiment by Frank et al. used feedback threshold of π/3 (“low precision” condition) and π/8 (“high precision” condition) and found no difference in absolute estimation error between these two conditions. This would be at odds with any model that predicts that encoding precision is higher in the “high precision” condition, which is what one may expect to happen in our model. However, it turns out that the predictions for this experiment are not that straightforward and that the model can actually account for the lack of an effect. The short explanation is that there is a threshold region in which the prediction barely changes as a function of threshold, due to the performance benefit of adding extra resource is almost exactly outdone by the added neural cost. For a more detailed explanation, we refer to the new Figure 6.

This point is now also discussed in a new Discussion section “Experimental predictions of incentive manipulations”.

4) Total precision vs. set-size: A novel and interesting prediction of the model is the non-monotonic relation between set-size and total precision (Figure 3B). Although the authors state that this point requires more empirical documentation, are the model results consistent with a non-monotonic encoding of target location that was reported by Balan et al. (2008) in monkey area LIP? That study found that, in a covert visual search task with different set sizes, the fidelity of target location encoding by area LIP was higher at set size 4 than at set size 2 or set size 6 (see Figure 5) – a non-linearity that was puzzling at the time but may gain new significance in light of this paper. Please add some discussion of this point to the manuscript.

We thank the reviewer for the reference, as we were not aware of that paper. However, after a careful study of the results reported in that paper, we don’t see how the non-monotonic trend in Figure 3 can be linked to the predicted non-monotonicity in the total amount of invested resource. The non-monotonicity in the Balan paper shows that the stimulus identity (target/distractor) can be decoded more accurately from neural data in N=4 trials compared to N=2 and N=6 trials. However, we do not see how decoding accuracy of a single item relates to the total amount of resource invested in all items. Although it would have been nice if the Balan paper backs up the non-monotonicity prediction, we believe that linking our prediction to their result would be a bit misleading, so we decided to not include this point. (However, if we misunderstood the reviewer’s suggestion, we would of course be happy to have another look at it after some clarification).

Nevertheless, the Balan paper is relevant to our work for other reasons and we now cite it at two different places in the Discussion. First, in the Discussion section about experimental predictions and, second, in the rewritten part about Neural mechanisms (see responses to previous comments).